# MINI-SEQUENCE TRANSFORMER: Optimizing Intermediate Memory for Long Sequences Training

**Cheng Luo**
California Institute of Technology
chengluo@caltech.edu

**Jiawei Zhao**
Meta FAIR
jwzhao@meta.com

**Zhuoming Chen**
Carnegie Mellon University
zhuominc@andrew.cmu.edu

**Beidi Chen**
Carnegie Mellon University
beidic@andrew.cmu.edu

**Anima Anandkumar**
California Institute of Technology
anima@caltech.edu

## Abstract

We introduce MINI-SEQUENCE TRANSFORMER (MST), a simple and effective methodology for highly efficient and accurate LLM training with extremely long sequences. MST partitions input sequences and iteratively processes mini-sequences to reduce intermediate memory usage. Integrated with activation recomputation, it enables significant memory savings in both forward and backward passes. In experiments with the Llama3-8B model, with MST, we measure no degradation in throughput or convergence even with 12x longer sequences than standard implementations. MST is fully general, implementation-agnostic, and requires minimal code changes to integrate with existing LLM training frameworks. Integrated with the huggingface library, MST successfully extends the maximum context length of Qwen, Mistral, and Gemma-2 by 12-24x.

## 1 Introduction

The development of Transformer [56] has been a remarkable journey, with each iteration pushing the boundaries of what is possible regarding model size, performance, and efficiency. One of the critical challenges in this journey has been managing the memory requirements of these models, particularly during training. As Transformers have significantly grown in size[10] and complexity [44], the memory demand has increased exponentially, necessitating innovative solutions to optimize memory usage while maintaining performance.

A significant milestone in this journey was the introduction of multi-query attention [50]. This technique dramatically reduced the size of the KV-cache during inference, which uses multiple query heads but single key and value heads. The idea was first adopted in the large-scale training of PaLM [12], then adopted and empirically tested in LLaMA [55]. As the field progressed, multi-query attention evolved into grouped query attention (GQA) [2], which relaxes the single key and value head restriction to multiple heads, and each head is coupled with a group of queries. It significantly improves the quality and is adopted by Llama2-70B [55] and Mistral-7B [24].

To further improve model quality, Llama3 [36] introduced a tokenizer with a vocabulary of 128K tokens, enabling more efficient language encoding than Llama2's 32K vocabulary. Additionally, Llama3 increased its MLP intermediate size from 11k to 14k. These changes reflect a trend toward more extensive vocabulary and intermediate sizes for better quality. Meanwhile, Llama3 maintains its hidden size of 4k for inference efficiency. This trend is also reflected in the Microsoft development of Phi-3 [1] compared with Phi-2 [23].

38th Conference on Neural Information Processing Systems (NeurIPS 2024).

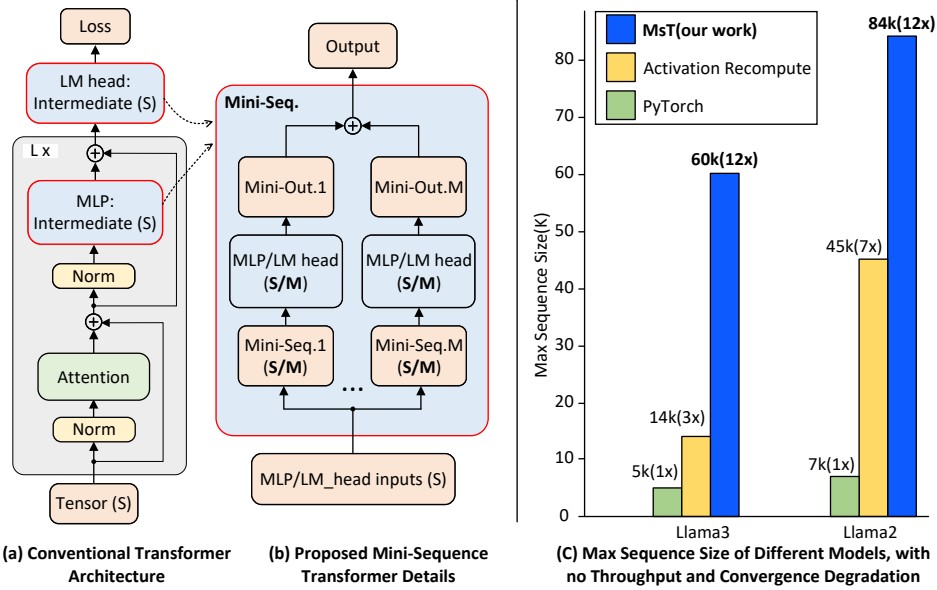

Figure 1: (a) Standard Transformer architecture. MLP's and LM-Head's activation sequence length is annotated with $S$. (b) MINI-SEQUENCE TRANSFORMER is used to replace MLP blocks and LM-Head block, which splits the input sequence $S$ into $M$ mini-sequences with sequence length $S/M$, where $M = 2$ on this figure. (c) Max sequence size for training Llama2/Llama3 on A100-80GB GPU, with no degradation of throughput or convergence using our approach.

These advancements have also brought about new memory challenges, particularly in the intermediate value of linear layers of multilayer perception (MLP) and language modeling head (LM-Head). The substantial increase in intermediate variables, which can be nearly ten times larger than the input variables, has severely limited the network's ability to expand sequence length and batch size. This limitation has made it difficult to train large models without restricting sequence length to 8K or relying on gradient accumulation or distributed systems to expand batch size.

**Our Approach:** Recognizing these challenges, we introduce MINI-SEQUENCE TRANSFORMER (MST), a simple and effective methodology for enabling highly efficient and highly accurate LLM training with extremely long sequence lengths by reducing intermediate memory overhead. MST introduces a per-layer mini-sequence where the input partitions work for each MLP and LM-Head block. MST partitions individual samples along the sequence dimension and iteratively processes each mini-sequence, combining all mini-sequence results to recover full-sequence outputs for these blocks. Our work also adopts activation recomputation [8]. We find no degradation in throughput or convergence even with sequences up to $12\times$ compared to a standard implementation of Llama3-8B, as shown in Figure 1(c).

To summarize, we make the following contributions to advance the long-sequence training:

- MST trains $12 - 24\times$ longer sequence lengths than existing systems on a single A100 GPU with no degradation in throughput and convergence of training.

- Fully general and implementation agnostic: MST supports most parameter-efficient training as it works independently with attention layers.

- Support for large-scale distributed training: MST works together with DeepSpeed-Ulysses [21] to support linear scaling sequence length by the number of GPUs.

- Easy-to-use and portable, requiring minimal code changes to the existing training frameworks like Huggingface [22]. The details can be referred to Appendix G.

In subsequent sections, we provide background and related work, a detailed discussion of MINI-SEQUENCE TRANSFORMER (MST) design, Hardware-efficient analysis, experimental evaluation, and comparison with existing work. This work is open-source under an MIT license on https://github.com/wdlctc/mini-s.

## 2 Background and Related Work

This section briefly overviews the performance characteristics of long sequence transformers on modern hardware(e.g., GPUs). We also describe some backgrounds of mini-batch training and activation recomputation, which inspire our work.

### 2.1 Transformer Architecture

Figure 1(a) is a sketch of the building blocks of a typical Transformer architecture [56]. It consists of input sequences $S$ sent into $L$ repeated block with attention and MLP, then computed output loss with LM-Head block. The inputs and outputs of each block are typically a 3D tensor of size $(B, S, d)$ where $B$ is micro batch size, $S$ is sequence length, and $d$ is hidden dimension. The intermediate value includes the $Q, K, V$ tensors of size $(B, S, d)$ within the attention block, the $I$ tensor of size $(B, S, I)$ within the MLP block, and the logits tensor of size $(B, S, V)$ of within LM-Head block. Here, $I$ represents the intermediate size of MLP, and $V$ represents the vocabulary size.

### 2.2 Hardware Performance of Long Sequence Training

**Memory Hierarchy.** GPUs have a memory hierarchy with larger but slower global GPU memory (high bandwidth memory; HBM) and smaller but faster-shared memory (SRAM). Transformers' high memory demand originates from the quadratic complexity of self-attention operations, where the memory needed to store attention scores for each token increases quadratically as the sequence length grows. This dramatic increase in memory demand can quickly overwhelm the capacity of the HBM, leading to OOM issues. Flashattention [15] uses kernel fusion to effectively mitigate the memory overheads associated with the quadratic growth in sequence length, and Xformer [41] deploys optimized memory access patterns that achieve linear memory scaling. Our work is partly inspired by memory optimization technologies, where our optimization targets are MLP and LM-Head.

**Occupancy.** GPUs have many threads executed in parallel; threads are grouped into thread blocks, which execute on streaming multiprocessors (SMs). Modern hardware has specialized units like tensor cores on NVIDIA GPU to accelerate mammals. In long sequence training scenarios where the sequence size tends to be long (>10k), parallelizing over the sequence dimension usually enables high GPU occupancy.

**Performance characteristics.** GPU operators can be classified as either compute-bound or memory-bound, which is determined by the time spent in arithmetic operations and the time spent accessing HBM. Typical self-attention with long sequence, MLP with the long intermediate size is a compute-bound operator because their core operators are matrix-multiply with a large inner dimension of sequence length. Then, cross-entropy with reduction is memory-bound.

### 2.3 Mini-Batch Training

Our work is inspired by Mini-Batch Training algorithms, also known as gradient accumulation. Mini-batch training algorithms [17, 40] can support large batch size by processing the training batch in smaller mini-batches, which allows the model to be trained on a subset of the data at a time, accumulating gradients over several mini-batch and only updating the parameter with accumulated gradient. This reduces the memory requirements compared to batch gradient descent [25], which enables training bigger batch sizes than GPU memory constrain. We are inspired by the idea and adapt it to train long sequences instead of large batch sizes.

### 2.4 Activation Recomputation

Activation recomputation [8], also known as gradient checkpointing, is a memory-saving technique for training large neural networks. This method trades computation for memory by discarding intermediate activations during the forward pass and recomputing them as needed during the backward pass. In standard training, all activations must be stored to compute gradients, which can lead to significant memory usage for large models or long sequences. Activation recomputation is orthogonal with our MST, and we integrate this method for better optimizing intermediate value. We analyze the memory efficiency of activation recomputation and its integration with MST on Sec 3.2.

# 3 MINI-SEQUENCE TRANSFORMER (MST): Algorithm, Analysis, and Distributed Extensions

We present our MINI-SEQUENCE TRANSFORMER (MST) mechanism to partition the input sequence into $M$ mini-sequences. We show how to compute the exact transformer block by gradient accumulation during the backward pass. Then, we analyze its memory efficiency and IO complexity, showing that our method is memory-efficient and throughput-equalized compared to the standard transformer. Based on the analysis, we found the optimal implementation of MST by selecting the best hyperparameters. We further show how MST can work on distributed settings by integrating with DeepSpeed [21].

We focus here on the forward pass for ease of exposition; Appendix B contains details for the backward.

## 3.1 Algorithms: Optimizing Intermediate Memory With Mini-Sequence Processing

Our idea arises from the observation of large intermediate values from transformer blocks. Given the inputs $X \in \mathbb{R}^{N \times d}$ in HBM, attention blocks and MLP blocks compute the output $O \in \mathbb{R}^{N \times d}$ and LM-head block computes the output $loss \in \mathbb{R}^1$, $N$ equals to sequence size $S$ here. We observe that the intermediate values are always larger than the input $X$ and output $O$, $loss$, illustrated in Table 1. Attention has intermediate values $\mathbf{Q}, \mathbf{K}, \mathbf{V} \in \mathbb{R}^{N \times d}$, which is $(1 + 2 \times d)/G$ larger than input size, where $(1 + 2 \times d/G = 1.5)$ in Llama3 setting. $G$ refers to the number of grouped query attention (GQA). MLP has intermediate value $I_{up}, I_{gate} \in \mathbb{R}^{N \times I}$, where $2 \times I/d = 7$ in Llama3 setting. LM-Head has $logits \in \mathbb{R}^{V \times d}$, where $V/d = 32$ in Llama3 setting. The detail setting of Llama3-8B is listed in Appendix C

Table 1: Intermediate value size analysis for transformer blocks

| Transformer Blocks | Input/Output Size | Peak Intermediate Value Size | Intermediate/Input Ratio [1] |
|---|---|---|---|
| Attention | $(B, S, d)/(B, S, d)$ | $(B, S, d) + 2 \times (B, S, d/G)$ | $(1 + 2 \times d/G) \approx 1.5$ |
| MLP | $(B, S, d)/(B, S, d)$ | $2 \times (B, S, I)$ | $(2 \times I)/d \approx 7$ |
| LM-Head | $(B, S, d)/1$ | $(B, S, V)$ | $V/d \approx 32$ |

[1] The ratio in Llama3 setting.

As flash attention and group query attention have minimized the intermediate value of attention, we put our focus on the MLP block and LM-Head block. Therefore, our implementation of MST is general enough to work with any attention: self-attention [56], cross-attention [5], causal attention [42], their sparse counterparts [11, 59, 48], and their various optimized kernels such as different versions of FlashAttention [15, 14]. Our implementation adopts FlashAttention2 [14] for the experiments.

**Input Partition.** We apply the mini-sequence technique to overcome the technical challenge of large intermediate values occupying HBM memory. We describe this in Algorithms 1, and 2, which represent MLP blocks and LM-Head from Llama serials. Their MLP block consists of three linear layers and SiLU function [46], and their LM-Head block consists of one linear layer and CrossEntropyLoss function[49]. The corresponding backward implementations can be referred to in Appendix B for more details. The main idea is to partition the input $X$ into mini-sequence $X_i$ as Algorithm 1 line 1 and Algorithm 2 line 1, then compute the output with respect to those mini-sequences. We get the exact same result as standard implementation by contacting all mini-sequence outputs.

**Gradient Accumulation.** One of our goals is to reduce intermediate values for backward passes. The backward pass typically requires the matrices $X \in \mathbb{R}^{N \times d}$, $I \in \mathbb{R}^{N \times I}$, $logits \in \mathbb{R}^{N \times V}$ to compute the gradients with respect to weights. However, by input partition the $X \in \mathbb{R}^{N_m \times d}$, we can reduce the intermediate value as $I \in \mathbb{R}^{N_m \times I}$, $logits \in \mathbb{R}^{N_m \times V}$ by $M \times$ in the backward pass in HBM. With gradient accumulation for all mini-sequences, all gradients are generated in the same way as standard implementation by introducing more memory loading time. However, as MLP is the standard computation-bound operator and LM-Head occupies only a small amount of total training time, MST would not affect the whole training speed with a significant reduction in memory overhead.

---

**Algorithm 1** Mini-Sequence MLP

---

**Require:** Matrices $X \in \mathbb{R}^{N \times d}$, MLP block, $W_{down}, \in \mathbb{R}^{I \times d}$, Weights of three linear layers $W_{gate}, W_{up} \in \mathbb{R}^{d \times I}, W_{down} \in \mathbb{R}^{I \times d}$
 1: Partition matrices $X$ into $M$ blocks $X_1, \ldots, X_m$ of size $N_m \times d$, where $N_m = N/M$
 2: **for** $1 \leq i \leq M$ **do**
 3:     Compute $\mathbf{O}'_i = MLP(X_i, W_{gate}, W_{up}, W_{down})$, $\mathbf{O}_i \in \mathbb{R}^{N_m \times d}$
 4: **end for**
 5: Contact $\mathbf{O} = \{\mathbf{O}'_i, \ldots, \mathbf{O}'_m\} \in \mathbb{R}^{N \times d}$
 6: Return $\mathbf{O}$.

---

**Algorithm 2** Mini-Sequence LM-Head

---

**Require:** Matrices $X \in \mathbb{R}^{N \times d}$, Labels $L \in \mathbb{R}^N$, Weights $W_{out} \in \mathbb{R}^{d \times V}$
 1: Partition matrices $X$ into $M$ blocks $X_1, \ldots, X_m$ of size $N_m \times d$, where $N_m = N/M$
 2: Partition labels $L$ into $M$ sub-label, $L_1, \ldots, L_m$ of size $N_m$, where $N_m = N/M$
 3: **for** $1 \leq i \leq M$ **do**
 4:     Compute $logits_i = X_i W_{out}$, $logits_i \in \mathbb{R}^{N_m \times V}$
 5:     Compute if $(i-1) * N_m \leq L_i \leq i * N_m$, $L_i = L_i$ else $L_i = -100$
 6:     Compute $loss_i = crossentropyloss(logits_i, L\_)$
 7: **end for**
 8: Compute $loss = \sum_1^M loss_i / M$
 9: Return $loss$.

---

### 3.2  Analysis: Memory Efficiency of MINI-SEQUENCE TRANSFORMER (MST)

We analyze the memory efficiency of MST. MST can reduce intermediate value by $M\times$ while maintaining the same throughput performance.

**Theorem 1.** *Let $S$ be the sequence length, $W_{mem}$ be the weight memory occupation, including weights, gradient, and optimizer. $A_{mem}$ be the activation memory occupation per sequence, $I_{mem}$ be the intermediate memory occupation per sequence. The peak memory of the standard transformer is achieved by $M = W_{mem} + S \times (I_{mem} + L \times A_{mem})$. Note that $L \times A_{mem} >> I_{mem}$ for standard transformer, as $A_{mem}$ lasts for all $L$ layers, but $I_{mem}$ only lasts for one layer.*

**Theorem 2.** *With OpenAI's activation recomputation[39], the $L \times A_{mem}$ could be reduced to $sqrt(L) \times A_{mem}$. Therefore the peak memory is reduced to $M = W_{mem} + S \times (I_{mem} + sqrt(L) \times A_{mem})$. For models with a large vocabulary and MLP intermediate, $sqrt(L) \times A_{mem} < I_{mem}$.*

**Theorem 3.** *MST can reduce intermediate value by $M\times$, so the memory occupation becomes $M = W_{mem} + S \times (I_{mem}/M + sqrt(L) \ times A_{mem})$. For GPU with maximum memory $M_{max}$, the maximum sequences length is contained by $S_{max} = \frac{(M_{max} - W_{mem})}{(I_{mem}/M + sqrt(L) \times A_{mem})}$. This sequence length would be much longer than the standard implementation with $S_{max} = \frac{(M_{max} - W_{mem})}{(I_{mem} + L \times A_{mem})}$.*

### 3.3  Analysis: IO Complexity and Memory of MINI-SEQUENCE TRANSFORMER (MST)

We analyze the IO complexity of MST, compared with consistent compute complexity, which can affect its compute-bound or memory-bound performance characteristics.

**Theorem 4.** *Let $S$ be the sequence length, $d$ be the hidden dimension, $I$ be the intermediate size, and $V$ be the voice size. Standard MLP returns $O = act((XW_{gate}) * (X_i W_{up})) * W_{down}$ with $O(SdI)$ FLOPS and MST MLP returns $O(SdI/M * M) = O(SdI)$ FLOPS. Standard LM-Loss returns $loss = crossentropyloss(XW, L)$ with $O(SdV + SV)$ FLOPS, and MST LM-Loss returns $O((SdV + SV)/M * M) = O(SdV + SV)$ FLOPS.*

**Theorem 5.** *Standard MLP requires $\Theta(Sd + SI + dI)$ HBM accesses, while MST (1) requires $\Theta(Sd + SI + dIM)$ HBM accesses. Standard LM-Head requires $\Theta(Sd + SV + dV)$ HBM accesses, while MST (2) requires $\Theta(Sd + SV + dVM)$ HBM accesses.*

For Llama3 values of $d$ (4096), $I$ (14336) and $V$ (128256), $SI, Sv$ is many time larger than $Sd$. For long sequence cases, the compute complexity and IO complexity are dominated by $SI$ and $SV$, where MST is close to standard implementation. However, for small sequence cases where $S << d$, the compute complexity and IO complexity are dominated by $dI$ and $dV$ while MST needs $dIM$ and $dVM$. Therefore, MST would cause throughput downgrades for small sequence lengths.

### 3.4 Chunk-based MINI-SEQUENCE TRANSFORMER (MST)

We present an optimized implementation of chunk-based MST designed to mitigate throughput reductions when training with small sequence data. The fundamental approach involves partitioning sequences $S$ into equally sized chunks of size $C$ (when possible), resulting in $M = S/C$ mini-sequences.

Our IO complexity analysis indicates that the number of mini-sequences $M$ influences the HBM accesses as $\Theta(Sd + SI + dIM)$ and $\Theta(Sd + SV + dVM)$. However, the HBM accesses remain stable at $\Theta(SI)$ and $\Theta(SV)$ provided that $dIM \leq SI$ and $dVM \leq SV$. It means $d \leq S/M$.

Therefore, by setting the chunk size to $C = S/M \geq d$, MST avoids throughput downgrades for small sequences. Intuitively, when the sequence size is smaller than the chunk size, MST does not split the input, thereby preventing any performance loss.

We apply chunk-based MST exclusively to MLP blocks by setting a constant chunk size $C$ equal to the hidden dimension, $C = d$. For LM-head blocks, we maintain a constant mini-sequence size of $M = V/d$, as these blocks contribute minimally to the overall training time of transformers.

### 3.5 Extension: Distributed MINI-SEQUENCE TRANSFORMER (MST)

We extend MINI-SEQUENCE TRANSFORMER (MST) to the distributed setting: we propose MST + SP, which can effectively scale the transformer using sequence parallelism(SP). In SP, the input tensor of each Transformer layer is divided along the sequence dimension, allowing for parallel computation across multiple GPUs. This segmentation, in conjunction with activation recomputation, results in a substantial reduction in activation memory requirements. It is worth noting that our proposed approach is orthogonal to most sequence parallelism, such as Megatron-LM [26], Deepspeed-Ulysses [21], Sequence parallelism [29], and Ring Attention [30]. Here, we take Deepspeed-Ulysses as an example of how they work together.

Figure 2 shows the design of extending MST with DeepSpeed-Ulysses. As with the transformers architecture, the design consists of an attention block with DeepSpeed-Ulysses, MLP, and LM-Head with MST's mini-sequence technology. The design consists of input sequences $S$ partitioned across available devices and mini-sequences. Each attention block Matrices $\mathbf{Q}, \mathbf{K}, \mathbf{V}$ are communicated through all-to-all collectives before and after the attention computation. The remaining modules of MLP and LM-Head use the sequence parallel and mini-sequence together. As DeepSpeed-Ulysses's main change is working on attention block and MST is working on MLP and LM-Head, it is straightforward to make them work together to scale sequence length.

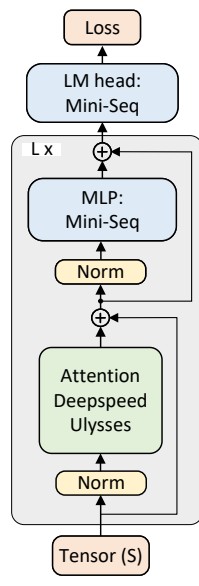

Figure 2: Distributed MINI-SEQUENCE TRANSFORMER.

## 4 Experiment

We evaluate the impact of using chunk-based MINI-SEQUENCE TRANSFORMER (MST) on Llama3[36], a state-of-the-art model for many NLP tasks. We also evaluate Qwen [6], Mistral [24], and Gemma-2 [54] for context length improvements. We validate our claims about scaling sequence length, reporting training time, and memory overhead. Distributed Extension results can be found in appendix E, which confirms that the sequence length of MST can scale linearly with the number of GPUs.

- **Maximun Sequence Length.** MST can train Llama3-8B with context length 60k and Llama3-7B with context length 84k on a single A100 GPU, outperforming the standard implementation by $12\times$. Also, it achieves $12 - 24\times$ than the standard implementation of Qwen, Mistral, and Gemma-2.

- **Training throughput.** MST maintains the same training throughput compared with standard long-sequence training. Moreover, the throughput can be slightly improved with a large batch size supported by MST.

## 4.1 Longer Sequence Length with MINI-SEQUENCE TRANSFORMER (MST)

**Llama3 and Llama2.** We train a Llama3-8B[36] MST and Llama2 models[43] MST by exploring the sequence length on a single A100 GPU with lossless training strategies, such as activation recomputation, fusing the backward operation with the optimizer update [34] and MST. Table 2 compares our maximum sequence and training time to the PyTorch standard implementation and Huggingface PEFT with activation recomputation. Our implementation trains $4\times$ longer sequence LLAMA-3 compared with activation recomputation and $12\times$ longer sequence compared with standard implementation. Also, our implementation trains $1.8\times$ longer sequence compared with activation recomputation and $12\times$ longer sequence compared with standard implementation.

Table 2: Maximum sequence length of Llama3-8B and Llama2-7B.

| Llama3-8B-hf Implementation | Maximum Sequence Length (K) |
|---|---|
| Llama3-8B-hf vanilla | 5 |
| Llama3-8B-hf activation recomputation | 14 |
| Llama3-8B-hf MST | 60 |
| Llama2-7B-hf vanilla | 7 |
| Llama2-7B-hf activation recomputation | 45 |
| Llama2-7B-hf MST | 84 |

**Qwen, Mistral, and Gemma-2.** We've extended our evaluation to include Mistral-7B, Qwen2-7B, and Gemma-2-9B, demonstrating significant increases in maximum sequence length ($12\times$ for Mistral-7B, $18\times$ for Qwen2-7B, $24\times$ for Gemma-2-9B) across these architectures. Among these models, MST provide best sequence extension for Gemma-2 of $24\times$. The critical observation here is that gemma-2 uses the largest vocal size (256k) than Mistral-7B (32k) and Qwen2(152k).

Table 3: Maximum sequence length of various models.

| Model Implementations | Maximum Sequence Length (K) |
|---|---|
| Mistral-7B vanilla | 5 |
| Mistral-7B activation recomputation | 42 |
| Mistral-7B MST | 70 |
| Qwen2-7B vanilla | 4 |
| Qwen2-7B activation recomputation | 13 |
| Qwen2-7B MST | 74 |
| gemma-2-9b vanilla | 1.5 |
| gemma-2-9b activation recomputation | 5 |
| gemma-2-9b MST | 36 |

**Combination with gradient accumulation.** Gradient Accumulation has been used during training Llama2 and Llama3, which helps them train larger batch sizes given limited available GPU memory. However, in Gradient Accumulation, instead of updating the model parameters after processing each batch of training data, the gradients are accumulated over multiple batches before updating. This means that the memory usage for gradients would occupy the memory used for activation. Therefore, using gradient accumulation during training would constrain the maximum sequence size.

Table 4 summarizes the maximum sequence length with gradient accumulation. The activation recomputation technology can train up to 8K sequences. Then MST can train up to 30k sequence length, which is $4\times$ longer sequence length than activation recomputation, and $21\times$ longer than vanilla. For Llama2-7B, MST can also train up to 55k sequence length.

Table 4: Maximum sequence length training with gradient accumulation.

| Model Implementation with gradient accumulation | Maximum Sequence Length (K) |
|---|---|
| Llama3-8B-hf vanilla | 1.5 |
| Llama3-8B-hf Activation Recomputation | 8 |
| Llama3-8B-hf MST | 32 |
| Llama2-7B-hf vanilla | 4 |
| Llama2-7B-hf activation recomputation | 38 |
| Llama2-7B-hf MST | 55 |

**Comparison and Combination with Lossy Methods.** We've comprehensively compared MST with quantization methods and the combinations between MST and quantization on Table 5. All lossy methods are HuggingFace official implementations. This comparison demonstrates MST's

superiority in enabling longer sequences for Llama3 training on a single A100 GPU. MsT alone (60K tokens) outperforms these lossy approaches (4bit 28k). When combined with quantization techniques, MsT achieves even more impressive results: MsT + 8-bit reaches 110K tokens (a 22× improvement over standard 8-bit), while MsT + 4-bit pushes the boundary to 140K tokens. We did not evaluate the effect of quantization on training loss.

Table 5: Maximum sequence length training with lossy method

| Llama3 Implementations | Maximum Sequence Length (K) |
|---|---|
| 8-bit | 5 |
| 4-bit | 10 |
| MST | 60 |
| MST + 8-bit | 110 |
| MST + 4-bit | 140 |

## 4.2 Faster Long Sequence Training with MINI-SEQUENCE TRANSFORMER (MsT)

We evaluate the training performance of MsT on Llama3-8B with 8k sequence and Llama2-7B with 4k sequence using a single A100 80G GPU. Table 6 compares the training time per step and TFLOPS achieved by MsT with the vanilla PyTorch implementation and activation recomputation technique.

Table 6: Training performance using MsT on single A100 80G GPU.

| Model Implementation | Batch Size | Training Time Per Step (s) | TFLOPS |
|---|---|---|---|
| Llama3-8B-hf vanilla | 1 | OOM | OOM |
| Llama3-8B-hf activation recomputation | 2 | 5.01 | 3271.42 |
| Llama3-8B-hf MsT | 2 | 5.13 | 3194.90 |
| Llama3-8B-hf MsT | 8 | 19.35 | 3386.13 |
| Llama2-7B-hf vanilla | 1 | 1.24 | 3290.88 |
| Llama2-7B-hf activation recomputation | 1 | 1.52 | 2684.67 |
| Llama2-7B-hf MsT without activation recomputation | 1 | 1.31 | 3115.03 |
| Llama2-7B-hf activation recomputation | 8 | 8.85 | 3703.48 |
| Llama2-7B-hf MsT | 8 | 9.33 | 3511.39 |
| Llama2-7B-hf MsT | 16 | 17.92 | 3656.17 |

For Llama3-8B, the vanilla implementation runs out of memory (OOM) with a batch size of 1. Activation recomputation allows training with a batch size of 2, achieving 3271.42 TFLOPS and a training time of 5.01 seconds per step. MsT, with the same batch size of 2, achieves a comparable 3194.90 TFLOPS with a slightly longer training time of 5.13 seconds per step. However, MsT's memory efficiency allows scaling the batch size to 8, resulting in an improved 3386.13 TFLOPS and a training time of 19.35 seconds per step.

In the case of Llama2-7B, the vanilla implementation can train with a batch size of 1, achieving 3290.88 TFLOPS and a training time of 1.24 seconds per step. For the same batch size, MsT without activation recomputation achieves 3115.03 TFLOPS with a training time of 1.31 seconds per step, demonstrating a 16% speedup over activation recomputation (2684.67 TFLOPS) and only a 5% slowdown compared to vanilla PyTorch. MsT further increases the batch size to 16, maintaining a similar 3656.17 TFLOPS with a training time of 17.92 seconds per step.

## 4.3 Better Models with Longer Sequences

**Language Modeling with Long Context.**  The memory efficiency of MsT allows us to increase the context length of llama by 4× than activation recomputation. Table 7 shows that training Llama3-8B with 30K context length achieved a 2.7× improvement in perplexity compared to the 8K baseline. We train a Llama3-8B[36] MsT on the LongAlpaca dataset[9]. The training lasts for two epochs and 10k steps for demonstration. For all implementation, we use the AdamW optimizer [32]. We use a weight decay of 0.001, gradient clipping of 1.0, and a constant learning rate of 1e-4. All batch sizes equal 16, with a gradient accumulation step of 16. The bf16 precision is also deployed.

Table 7: LLAMA3-8b with MsT, with $4times$ larger context length compared to activation recomputation.

| Llama3-8B-hf Implementation | Context length | LongAlpaca-12k (ppl) | loss | Training Time |
|---|---|---|---|---|
| Activation Recomputation | 8k | 9.34 | 2.23 | 25.6 hours |
| MsT | 8k | 7.41 | 2.00 | 26.5 hours |
| MsT | 16k | 3.53 | 1.26 | 62.5 hours |
| MsT | 30k | 3.45 | 1.23 | 233 hours |

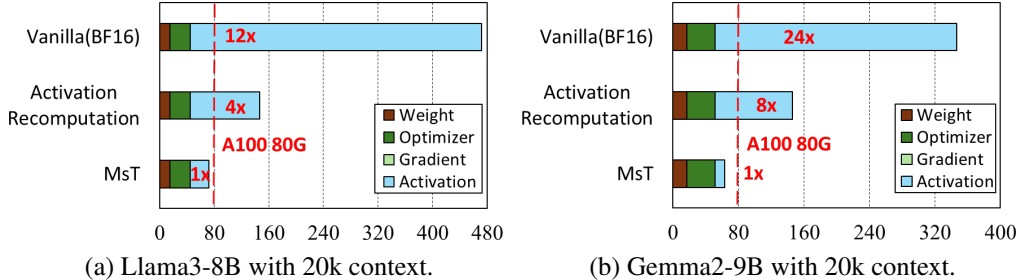

|                         | (a) Llama3-8B with 20k context. | (b) Gemma2-9B with 20k context. |
|:---:|:---:|:---:|

Figure 3: Memory consumption of pre-training Llama3-8B and Gemma2-9B models with a batch size of 1 on a single A100 device, with activation recomputation and MST. Note that long-sequence training gradients overlap with activation, so gradients are not shown in bars.

## 5 Ablation Study:

### 5.1 Memory Optimization of MINI-SEQUENCE TRANSFORMER (MST)

MST introduces a series of memory optimizations to reduce the memory overhead of long-sequence training. To understand the effectiveness of MST memory optimizations, we perform an ablation study that incrementally turns off these optimizations (mini-sequence, activation recomputation) and measures the memory requirements. We consider three options: vanilla (standard Pytorch with BF16), activation recomputation only, and MST with activation recomputation.

Figure 3 shows the results. We analyze the peak memory usage of Llama3-8B and Gemma2-9B, with a sequence length of 20k. For sequence length 20k of Llama3-8B and Gemma2-9B, only MST can make the model fit into A100 GPU. The rest of the memory consumption is estimated based on its model architectures and theoretical activation amount. For Llama3, activation recomputation can reduce the memory overhead of activation by $3\times$, and MST can further reduce $4\times$ memory overhead based on activation recomputation. For Gemma2-9B, MST achieves $24\times$ longer sequence than vanilla and $8\times$ longer sequence than activation recomputation. This improvement from $12\times$ to $24\times$ is due to Gemma2-9B's higher intermediate/input ratio (8 for MLP and 72 for the LM head) compared to Llama3 (7 for MLP and 32 for the LM head, as shown in Table 1). Further details on the memory ablation study can be found in Appendix D.

### 5.2 How many mini-sequences are needed during training

We observe that increasing $M$, the number of mini-sequences, can enhance memory efficiency; however, this enhancement has a certain upper limit. Specifically, increasing $M$ can also affect throughput performance. Appendix F provides details regarding these limitations and their effects. This observation allows us to identify the optimal configuration for memory optimization and achieve the best balance between memory performance, consistent with our analysis in Sec 3.2 and 3.3.

We found that the best balance for memory and throughput is achieved by the optimal values of $C$ for chunk-based MLP $C = d, M = S/d$, where $d$ is the hidden size. For the LM-Head, the original MST is employed for memory saving, and the optimal setting for $M$ is determined by $M = V/d$, specifically 32 for Llama3 and 64 for Gemma-2. This value provides the best memory efficiency.

## 6 Limitations and Future Directions

We discuss the limitations and future directions. Related work is also given in Appendix A.

**Compiling to CUDA.** Our current approaches are built on Pytorch implementation. This may constrain performance and low-level memory savings. It can be improved by fused kernel and cuda optimization, which can be our next step.

**Combination with memory optimization.** Our goal is to increase sequence length while maintaining performance and accuracy. Relaxing these requirements, MST can be combined with activation offload to extend sequence length as $S_{max} = \frac{(M_{max} - W_{mem})}{(I_{mem}/M + A_{mem})}$, or with quantization to extend sequence length as $S_{max} = \frac{bf16}{4bit/8bit} \frac{(M_{max} - W_{mem})}{(I_{mem}/M + L \times A_{mem})}$. This combination can be explored in future research.

## Acknowledgements

We thank Vast AI for computational resources renting.

A. Anandkumar is supported by the Bren named chair professorship, Schmidt AI 2050 senior fellowship, ONR (MURI grant N00014-18-12624).

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

## A  Related Work

**Long Sequences Model.**    The ability to train large models with long sequences is becoming increasingly important across various domains, from generative AI to scientific discovery. In generative AI, tasks such as conversational AI, knowledge-rich long document summarization, and video generation demand reasoning over extended contexts in both spatial and temporal dimensions. Multimodal foundation models process speech, images, and waveforms simultaneously, requiring long context reasoning over high-dimensional inputs with lengthy sequences. Similarly, chapter and book-level summarization, estimated to involve tens to hundreds of thousands of words, hold great importance in conversational AI and abstractive summarization tasks [7, 28, 33] and have demonstrated benefits from long sequence training [58, 47, 36].

The emergence of ChatGPT and subsequent open-source and commercial large language models has propelled chat applications to the forefront of modern AI, making them more relevant than ever. Efficiently processing long sequences is vital for supporting more extensive conversation histories in these applications [55]. Long sequence capability is equally important for AI in scientific fields, enabling better understanding and advancements in healthcare [31], climate and weather forecasting [38], and large-scale molecular simulations [62].

**Lossy Long Sequence Training.**    One direction is making LLM able to process arbitrarily long sequences efficiently by sacrificing the perception window of the network. Sliding window attention is introduced [13] to handle infinitely long sequences as input. However, it disregards information beyond the effective receptive field. Longformer [7] extends this idea, which caches on a block-by-block basis, which increases the perception window size, so as TransformerFAM [20]. StreamLLM [57] is not constrained by a given window but selectively disregards information between the first token and given windows. They struggle to capture long-range dependencies beyond this fixed range, but the approximation quality also seems to degrade at long sequence lengths. MsT can work directly with them to increase the window size for better quality.

**Memory Efficient Training.**    As the demand for long sequence processing continues to grow across various domains, developing efficient methods for training large models with extended context becomes increasingly essential for advancing the state of the art in AI. Parameter-efficient fine-tuning (PEFT) methods aim to reduce the memory footprint and computational related to parameters and gradients. Adafactor [51] achieves sub-linear memory cost by factorizing the second-order statistics using a row-column outer product. Low-Rank Adaptation (LoRA) [18] reduces the memory footprint of pre-trained models using low-rank adaptors with a low-rank weight adaptor for each layer. Several variants of LoRA have been proposed to enhance its performance. [52, 60]. Quantization is another widely used technique in PEFT to reduce the memory cost of optimizer states [16].

Other techniques focus on reducing the memory footprint and computational related to activation. Gradient accumulation is a method that allows for training larger batch sizes by accumulating gradients over multiple mini-batches before performing an optimization step [40]. This approach enables training with larger effective batch sizes while maintaining a smaller physical batch size, reducing activation memory requirements. A similar work of activation offloading [45] moves the checkpointed activations to the CPU asynchronously and prefetches the offloaded activations back from the CPU during backward. There are related works in sparse Transformers, mainly focusing on full-attention approximation, such as sparse attention [11, 59]. Recent works have also focused on single-GPU memory and compute-efficient attention. A popular example in this category is Flash attention [15], which leverages known techniques such as tiling and recomputation for computing and memory efficiency. Also, some work put their interest in cross-entropy. FlashCE [3] optimizes cross-entropy by leveraging sparse data structures and CUDA optimizations to enhance speed and memory efficiency. Efficient cross-entropy [35] introduces a memory-efficient variant of cross-entropy loss to reduce activation memory by storing only essential computations. These works are orthogonal to our work and can be leveraged accordingly to further improve the efficiency of Transformer-based models.

**Distributed Training.** Distributed training techniques have become essential for training large language models (LLMs) due to their immense computational and memory requirements. By splitting workloads across multiple GPUs, these methods help alleviate memory bottlenecks and enable the training of models that would otherwise be infeasible on a single device. Data parallelism [27] replicates the model on multiple devices, processing different data batches in parallel and synchronizing gradients across GPUs. Tensor parallelism [53] divides individual layers of the model across GPUs, enabling more efficient memory usage for extremely large models. Another prominent method is fully sharded data parallelism (FSDP) [61], which extends the data parallel approach by sharding both model parameters and optimizer states across devices, thus further reducing memory overhead. Sequence parallelism [26, 21, 29, 30] specializes in optimizing memory usage for transformer-based models by partitioning sequences across GPUs and reducing activation memory. In addition to these, pipeline parallelism [19] splits the model into segments, with each segment assigned to a different GPU, and processes data in a pipeline fashion, improving efficiency by overlapping computation and communication. Hybrid parallelism [37] combines data, tensor, and pipeline parallelism to maximize resource utilization depending on the model's architecture and available hardware.

# B   Algorithm Details

We describe the full details of MINI-SEQUENCE TRANSFORMER (MST) backward pass. Algorithm 3 shows the MLP backward, and Algorithm 4 shows the LM-Head backward.

---

**Algorithm 3** Mini-Sequence MLP Backward

---

**Require:** Gradients of output $\nabla O \in \mathbb{R}^{N \times d}$, Matrices $X \in \mathbb{R}^{N \times d}$, Weights of three linear layers $W_{gate}, W_{up} \in \mathbb{R}^{d \times I}, W_{down} \in \mathbb{R}^{I \times d}$
 1: Partition matrices $X$ into $M$ blocks $X_1, \ldots, X_m$ of size $N_m \times d$, where $N_m = N/M$
 2: Partition matrices $\nabla O$ into $M$ blocks $\nabla O_1, \ldots, \nabla O_m$ of size $N_m \times d$, where $N_m = N/M$
 3: **for** $1 \le i \le M$ **do**
 4:     Compute $\nabla X_i = \nabla MLP(\nabla O_i)$
 5:     Compute $\nabla W_{down}, \nabla W_{up}, \nabla W_{gate} += \nabla MLP_{gradient}(\nabla O_i, X_i)$
 6: **end for**
 7: Concatenate $\nabla X = \nabla X_1, \ldots, \nabla X_m \in \mathbb{R}^{N \times d}$
 8: Return $\nabla X, \nabla W_{gate}, \nabla W_{up}, \nabla W_{down}$.

---

We now observe about MST backward pass that when computing the gradients of MLP and LM-Head, we do not need to use full input and intermediates data. Instead, we can use $1/M$ reduced data with mini-sequence, significantly reducing the intermediate value memory overhead.

The main idea of backward is to accumulate gradients $\nabla W$ generated from each mini-sequence $X_i$ as Algorithm 3 line 5 and Algorithm 4 line 8. We get the exact same result as standard implementation by accumulating all mini-sequence gradients.

**Algorithm 4** Mini-Sequence LM-Head Backward

---

**Require:** Loss gradients $\nabla loss \in \mathbb{R}^1$, Logits $\in \mathbb{R}^{N \times V}$, Labels $L \in \mathbb{R}^N$, Weights $W_{out} \in \mathbb{R}^{d \times V}$
1: Partition matrices $X$ into $M$ blocks $X_1, \ldots, X_m$ of size $N_m \times d$, where $N_m = N/M$
2: Partition labels $L$ into $M$ sub-labels $L_1, \ldots, L_m$ of size $\frac{N}{m}$, where $\frac{N}{m} = \frac{N}{M}$
3: Activation Recomputation with backward
4: **for** $1 \leq i \leq M$ **do**
5:     Compute $logits_i = X_i W_{out}, logits_i \in \mathbb{R}^{N_m \times V}$
6:     Compute $\nabla logits_i = \text{CrossEntropyLossBackward}(Logits_i, L_i)$
7:     Compute $\nabla X_i = \nabla logits_i W_{out}^T, \nabla X_i \in \mathbb{R}^{\frac{N}{m} \times d}$
8:     Compute $\nabla W_{out} += X_i^T \nabla logits_i$
9:     Compute $\nabla X_i = \nabla X_i \odot \nabla loss$
10:    Compute $\nabla W_{out} = \nabla W_{out} \odot \nabla loss$
11: **end for**
12: Concatenate $\nabla X = \nabla X_1, \ldots, \nabla X_m \in \mathbb{R}^{N \times d}$
13: Return $\nabla X, \nabla W_{out}$.

---

## C  Llama2 and Llama3: Model Architecture Comparison

This appendix highlights the key architectural differences between the Llama2-7B and Llama3-8B models implemented by Hugging Face. The main distinction lies in the configuration of the MLP blocks and LM-Head (linear with cross-entropy loss) within the model architecture.

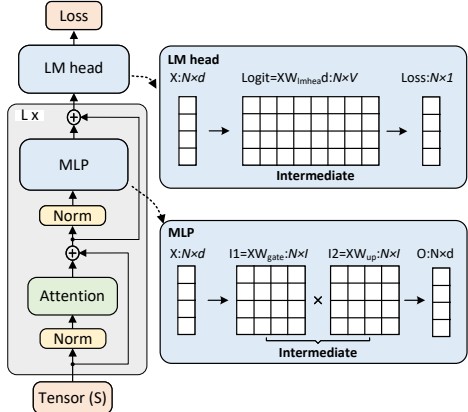

Figure 4: Standard Transformer architecture with the highlight of LM-head and MLP.

Figure 4 illustrates a standard Transformer model's architecture, focusing on the MLP and LM-head components. The intermediate tensors (I1, I2) have larger dimensions than the input and output. Also, the logit tensor has a much larger vocabulary dimension (V) than the hidden dimension (d).

Figure 5: Model Architecture of HuggingFace Implementation of Llama2-7B

```
LlamaForCausalLM(
  (model): LlamaModel(
    (embed_tokens): Embedding(32000, 4096)
    (layers): ModuleList(
      (0-31): 32 x LlamaDecoderLayer(
        (self_attn): LlamaFlashAttention2(
          (q_proj): Linear(in_features=4096, out_features=4096, bias=False)
          (k_proj): Linear(in_features=4096, out_features=4096, bias=False)
          (v_proj): Linear(in_features=4096, out_features=4096, bias=False)
          (o_proj): Linear(in_features=4096, out_features=4096, bias=False)
          (rotary_emb): LlamaRotaryEmbedding()
        )
        (mlp): LlamaMLP(
          (gate_proj): Linear(in_features=4096, out_features=11008, bias=False)
          (up_proj): Linear(in_features=4096, out_features=11008, bias=False)
          (down_proj): Linear(in_features=11008, out_features=4096, bias=False)
          (act_fn): SiLU()
        )
        (input_layernorm): LlamaRMSNorm((4096,), eps=1e-05)
        (post_attention_layernorm): LlamaRMSNorm((4096,), eps=1e-05)
```

```
      )
    )
    (norm): LlamaRMSNorm((4096,), eps=1e-05)
    (rotary_emb): LlamaRotaryEmbedding()
  )
  (lm_head): Linear(in_features=4096, out_features=32000, bias=False)
)
```

**Llama2-7B Model Architecture:**    As shown in Figure 5, the Llama2-7B model employs an MLP block configuration with the following characteristics:

- MLP Block: The first linear layer projects the input from a hidden size of 4096 to an intermediate size of 11008. The second linear layer projects the intermediate representation from 11008 to 4096, the hidden size.

- LM-Head (Output Projection with Linear Loss): The LM-Head in Llama2-7B consists of a linear layer that projects the hidden representation from a size of 4096 to a vocabulary size of 32000. The output of the linear layer is then passed through a cross-entropy loss function to compute the training loss.

Figure 6: Model Architecture of HuggingFace Implementation of Llama3-8B.

```
LlamaForCausalLM(
  (model): LlamaModel(
    (embed_tokens): Embedding(128256, 4096)
    (layers): ModuleList(
      (0-31): 32 x LlamaDecoderLayer(
        (self_attn): LlamaFlashAttention2(
          (q_proj): Linear(in_features=4096, out_features=4096, bias=False)
          (k_proj): Linear(in_features=4096, out_features=1024, bias=False)
          (v_proj): Linear(in_features=4096, out_features=1024, bias=False)
          (o_proj): Linear(in_features=4096, out_features=4096, bias=False)
          (rotary_emb): LlamaRotaryEmbedding()
        )
        (mlp): LlamaMLP(
          (gate_proj): Linear(in_features=4096, out_features=14336, bias=False)
          (up_proj): Linear(in_features=4096, out_features=14336, bias=False)
          (down_proj): Linear(in_features=14336, out_features=4096, bias=False)
          (act_fn): SiLU()
        )
        (input_layernorm): LlamaRMSNorm((4096,), eps=1e-05)
        (post_attention_layernorm): LlamaRMSNorm((4096,), eps=1e-05)
      )
    )
    (norm): LlamaRMSNorm((4096,), eps=1e-05)
    (rotary_emb): LlamaRotaryEmbedding()
  )
  (lm_head): Linear(in_features=4096, out_features=128256, bias=False)
)
```

**Llama3-8B Model Architecture:**    As shown in Figure 6, the Llama3-8B model employs an MLP block configuration with the following characteristics:

- MLP Block: The first linear layer projects the input from a hidden size of 4096 to a larger intermediate size of 13824. The second linear layer then projects the intermediate representation from a size of 13824 back to the hidden size of 4096.

- LM-Head (Linear with Cross-Entropy Loss): In Llama3-8B, the LM-Head also consists of a linear layer, but it projects the hidden representation from a size of 4096 to a larger vocabulary size of 32000. Like Llama2-7B, the output of the linear layer is passed through a cross-entropy loss function to compute the training loss.

The increased intermediate size in the MLP block of Llama3-8B allows the model to capture complex patterns and transformations more effectively. Additionally, the larger vocabulary size in the LM-HEAD of Llama3-8B enables the model to generate more diverse and nuanced outputs.

It is worth noting that while the dimensions of the MLP block and LM-Head differ between Llama2-7B and Llama3-8B, the overall structure and functionality of these components remain the same. The cross-entropy loss function in the LM-Head measures the discrepancy between the predicted word probabilities and the target words during training, guiding the model to generate more accurate and contextually relevant outputs. These architectural differences contribute to Llama3-8B's enhanced performance and capabilities compared to its predecessor, Llama2-7B, while maintaining a consistent overall structure. However, the unexpected large intermediate value also comes from the change of MLP blocks and LM-HEAD (linear with cross-entropy loss), which we will discuss in the following section.

Moreover, this trend is also reflected in the Microsoft development of Phi-3 [1] compared with Phi-2 [23]. Whose vocabulary size increased from 50k to 100K (2×), intermediate size increased from 10k to 16k (1.6×), and hidden size slightly increased from 2560 to 3072 (1.2×).

In conclusion, these models share an obvious trend: the ratio between intermediate and hidden size (also vocabulary and hidden size) is becoming larger.

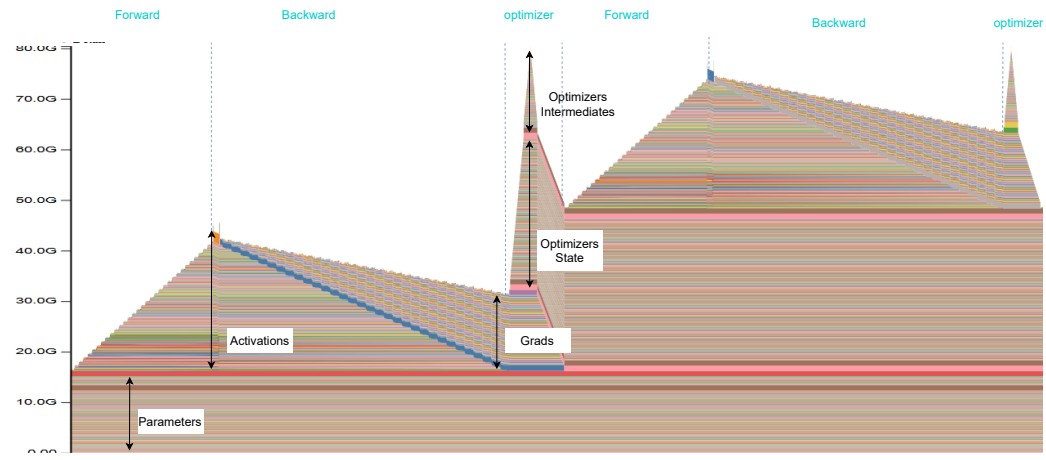

Figure 7: Memory Visualizaion of training Llama3 8B with 4k sequence length on A100-80G.

## D   MINI-SEQUENCE TRANSFORMER's Memory Optimization Detials

We compare MINI-SEQUENCE TRANSFORMER (MST) with capturing and visualizing memory snapshots. We take vanilla Pytorch training for Llama3-8B with 4k sequence length as an example to show how the memory changes with the timeline.

**vanilla.**   Figure 7 shows the vanilla example. The model parameters had already been loaded into memory before the training step, so we immediately see a chunk of memory devoted to the weights. For Llama3-8B, its weight would be 15GB. As we start our forward pass, memory is allocated gradually for the activations or the tensors we are saving to be able to compute gradients in the backward pass. Here, the memory allocated for activation is larger than the weight, with around 29GB. Once we start the backward pass, the activations are gradually freed while the memory of the gradients starts building up. As the gradient is equal to the size of weights, which is smaller than activation, we can observe the memory usage drop to around 30 GB. Lastly, as the optimizer kicks in, its state will be lazily initialized, so we should see the optimizer state memory gradually increase during the optimizer step of the first training loop only. In future loops, the optimizer memory will remain and be updated. The memory for the gradients is then freed accordingly at the end of every training loop when it is called zero grade. Here, the optimizer would take 2× of weights when using Adam with 30GB, and the optimizer intermediate is equal to the size of weight with 15. Therefore, the peak memory usage is during the optimizer step, which equals the sum size of weight, gradient, optimizer state, and optimizer intermediates, which roughly equals to 5× of weight size with 75GB as shown in Table 8.

Table 8: Memory overhead of training Llama3-8B on single A100 80G GPU.

| Llama3-8B-hf | Memory Overhead | Within Peak Memory |
|---|---|---|
| Activation | 29 | 0 |
| Weight | 15 | 15 |
| Gradient | 15 | 15 |
| Optimizer | 45 | 45 |
| Total | - | 75 |

**optimizer-in-backward.**   The first memory optimization discussed here is optimizer-in-backward [4]. It fuses the optimizer state with a backward pass to save the memory of the gradient and optimizer intermediate. The memory visualization of optimizer-in-backward is shown in Figure 8, where there

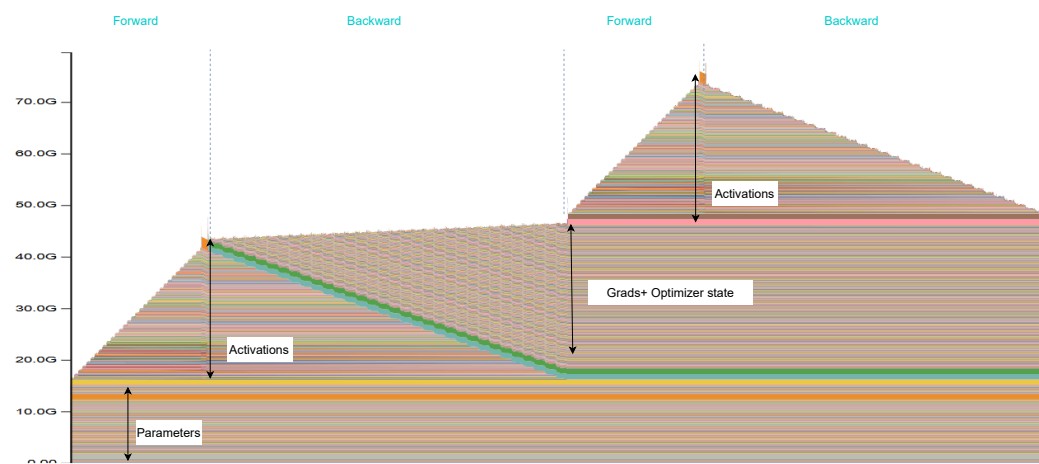

Figure 8: Memory Visualizaion of training Llama3 8B with 4k sequence length on A100-80G. The optimizer in the Backward technique is deployed here

is no optimizer stage but only forward and backward state. The backward time would become larger as a result of fusion. Using this technology, the peak memory would change into the sum of weight, optimizer state, and activations. Although we successfully saved 30GB of memory overhead of gradient and optimizer intermediates, it adds up to 29GB memory overhead of activations, with only 1GB of memory saving. Totally it consumes 74GB of memory, as shown in Table 9. It would be worse if sequence length were increased, introducing more activation into LLM training. Therefore, optimizer-in-backward can hardly benefit long sequence training, but it simplifies the training process, so we would include this technique in the following discussion.

Table 9: Memory overhead of training Llama3-8B on single A100 80G GPU. The optimizer in the Backward technique is deployed.

| Llama3-8B-hf | Memory Overhead | Within Peak Memory |
|---|---|---|
| Activation | 29 | 29 |
| Weight | 15 | 15 |
| Gradient | 15 | 0 |
| Optimizer | 30 | 30 |
| Total | - | 74 |

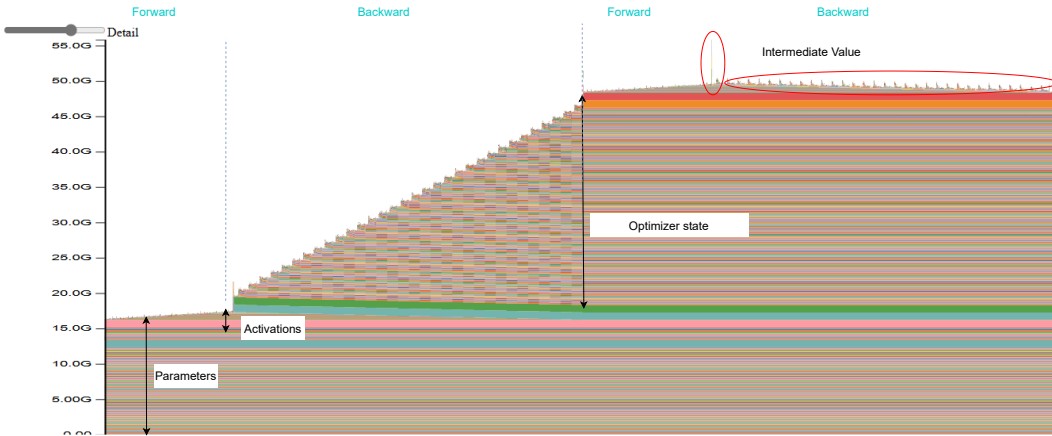

Figure 9: Memory Visualizaion of training Llama3 8B with 4k sequence length on A100-80G. Activation Recomputation technique is deployed here

**Activation Recomputation.** Activation Recomputation is a powerful technique used in our analysis. It can significantly reduce the activation at the cost of a small decrease in the training speed due

to recomputing parts of the graph during back-propagation. As shown in figure 9, it successfully reduces the total memory overhead from 74GB to 52GB and reduces activation memory overhead from 29GB to 7GB with $4\times$ memory saving. However, we can easily find many singular points in the graph, which appear as an impulse signal. This impulse signal's duration is concise, meaning that it is intermediate data that is briefly created in the forward/backward pass and immediately removed from the GPU HBM memory. The most prominent intermediate data is several times the total activation (4-5 times in our data analysis). These intermediate data seriously affect the training performance of long sequences and become the activation bottleneck of the row.

Table 10: Memory overhead of training Llama3-8B on single A100 80G GPU. Activation Recomputation technique is deployed.

| Llama3-8B-hf | Memory Overhead | Within Peak Memory |
|:---:|:---:|:---:|
| Activation | 7 | 7 |
| Weight | 15 | 15 |
| Gradient | 15 | 0 |
| Optimizer | 30 | 30 |
| Total | - | 52 |

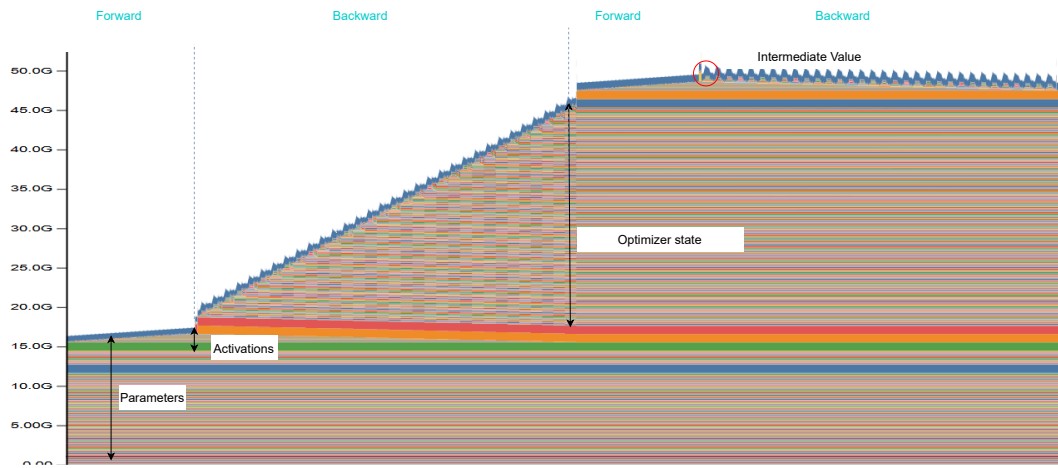

Figure 10: Memory Visualizaion of training Llama3 8B with 4k sequence length on A100-80G. MINI-SEQUENCE TRANSFORMER technique is deployed here

**MINI-SEQUENCE TRANSFORMER (MST)**    Inspired by the observation from Activation Recomputation techniques, we propose MST to reduce the intermediate value during training. We successfully decrease the memory overhead of activation from 7GB to 4GB, while the intermediate value is significantly reduced. This is because only $1/M$ intermediate values are used for computing the forward outputs, backward errors, and gradients during both the forward and backward.

Table 11: Memory overhead of training Llama3-8B on single A100 80G GPU. MINI-SEQUENCE TRANSFORMER technique is deployed.

| Llama3-8B-hf | Memory Overhead | Within Peak Memory |
|:---:|:---:|:---:|
| Activation | 2 | 2 |
| Weight | 15 | 15 |
| Gradient | 15 | 0 |
| Optimizer | 30 | 30 |
| Total | - | 48 |

**Conclusion.**    we compare the memory usage over time when training the Llama3-8B model using the standard transformer architecture versus using MST, which is shown on 11.

In Figure 11(a), which shows the memory timeline for the standard Llama3-8B training, we can see that the memory usage is dominated by three main components: the model weights (in blue), the optimizer state (in green), and the intermediate memory (highlighted by the red circle). The peak memory usage reaches around 67GB.

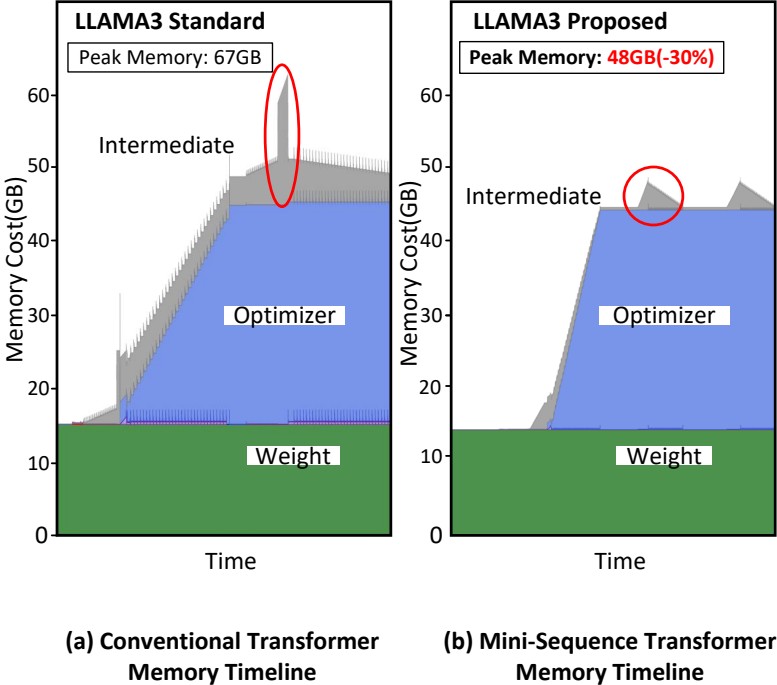

**(a) Conventional Transformer
Memory Timeline**

**(b) Mini-Sequence Transformer
Memory Timeline**

Figure 11: Memory Visualization. (a) The memory timeline of training Llama3-8B using standard transformer architecture, Red cycle highlights the intermediate memory (b) The memory timeline of training Llama3-8B using MsT, Red cycle highlights the intermediate memory has been narrowed.

In contrast, Figure 11(b) demonstrates the memory timeline when training Llama3-8B with MsT. The critical difference is that the intermediate memory, again highlighted by the red circle, has been significantly reduced or "narrowed" compared to the standard transformer case. As a result, the peak memory usage with MsT is around 47GB, achieving a 30% reduction compared to the standard transformer.

The memory timelines illustrate that MsT effectively reduces the intermediate memory footprint during training, significantly contributing to overall memory consumption. By minimizing the intermediate memory, MsT enables more efficient memory utilization and allows training with longer sequence lengths or larger batch sizes while staying within the hardware's available memory limits.

## E    Scaling to Extreme Long Sequence on Distributed Setting

We evaluate our distributed extension of MsT on Llama3-8B Llama2-7B models and compare against vanilla DeepSpeed-Ulysses's sequence parallelism on 2,4,8 GPUs, respectively, for the max sequence length and corresponding training time. The results of this evaluation are shown in Table 12.

Table 12: Maximum sequence length of Llama3-8B, running on distributed setting.

| Model Implementation | GPU numbers | | |
|---|---|---|---|
| | 2 | 4 | 8 |
| Llama3-8b-hf MsT | 120 | 240 | 480 |
| Llama2-7B-hf MsT | 160 | 320 | 640 |

## F    How many mini-sequences are needed during pre-training

We also provide insights into the performance characteristics and memory usage of the Llama3-8B model when using the MINI-SEQUENCE TRANSFORMER (MsT) approach with different numbers of mini-sequences (M) and sequence lengths.

Table 13 shows the execution time for different sequence lengths and mini-sequence settings. As the number of mini-sequences (M) increases, the execution time slightly increases, especially for shorter

sequences. However, for longer sequences (e.g., 80000), the execution time remains relatively stable across different mini-sequences (M).

Table 13: LM-head time for different sequence lengths and mini-sequence settings

| LM-head time | 1024 | 2048 | 4096 | 8192 | 20000 | 40000 | 80000 |
|---|---|---|---|---|---|---|---|
| standard | 0.01 | 0.02 | 0.04 | 0.09 | 0.2 | 0.4 | 0.85 |
| M=2 | 0.02 | 0.04 | 0.07 | 0.14 | 0.31 | 0.67 | 1.36 |
| M=4 | 0.03 | 0.04 | 0.07 | 0.14 | 0.33 | 0.67 | 1.33 |
| M=8 | 0.04 | 0.05 | 0.08 | 0.14 | 0.34 | 0.67 | 1.34 |
| M=16 | 0.06 | 0.07 | 0.09 | 0.16 | 0.34 | 0.68 | 1.35 |
| M=32 | 0.11 | 0.11 | 0.14 | 0.19 | 0.37 | 0.69 | 1.36 |

Table 14 shows the memory usage in gigabytes (GB) for different sequence lengths and mini-sequence settings for the LM-Head component. The standard setting without mini-sequences consumes 59.92 GB for a sequence length of 80000. By increasing the number of mini-sequences, memory usage decreases significantly. With M=16, the memory usage reduces to 9.12 GB for the same sequence length, achieving an 84.8% reduction in memory consumption. This is further improved with M=32 to 89.8%.

Table 14: LM-head memory usage (in GB) for different sequence lengths and mini-sequence settings

| LM-head memory | 1024 | 2048 | 4096 | 8192 | 20000 | 40000 | 80000 |
|---|---|---|---|---|---|---|---|
| standard | 3.20 | 3.46 | 4.94 | 7.91 | 16.46 | 30.95 | 59.92 |
| mini-seq=2 | 2.59 | 3.21 | 4.46 | 6.95 | 14.14 | 26.31 | 50.66 |
| mini-seq=4 | 2.28 | 2.60 | 3.24 | 4.52 | 8.20 | 14.44 | 26.92 |
| mini-seq=8 | 2.13 | 2.30 | 2.63 | 3.30 | 5.24 | 8.51 | 15.05 |
| mini-seq=16 | 2.06 | 2.15 | 2.33 | 2.70 | 4.14 | 5.54 | 9.12 |
| mini-seq=32 | 2.02 | 2.07 | 2.18 | 2.39 | 3.01 | 4.06 | 6.15 |

Table 15 shows the execution time for different sequence lengths and mini-sequence settings. Like the LM-Head, increasing M leads to slightly longer execution times, particularly for shorter sequences. However, the impact on execution time is minimal for longer sequences (e.g., 80000).

Table 15: MLP time (in seconds) for different sequence lengths and mini-sequence settings

| MLP time | 1024 | 2048 | 4096 | 8192 | 20000 | 40000 | 80000 |
|---|---|---|---|---|---|---|---|
| standard | 0.05 | 0.08 | 0.16 | 0.31 | 0.74 | 1.52 | 2.96 |
| M=2 | 0.05 | 0.10 | 0.17 | 0.32 | 0.76 | 1.49 | 3.05 |
| M=4 | 0.07 | 0.11 | 0.19 | 0.33 | 0.79 | 1.52 | 2.99 |
| M=8 | 0.12 | 0.15 | 0.22 | 0.38 | 0.81 | 1.58 | 3.05 |

For the MLP component, Table 16 demonstrates the memory usage for different sequence lengths and mini-sequence settings. The standard setting consumes 14.72 GB for a sequence length of 80000, while using M=8 mini-sequences reduces the memory usage to 11.66 GB, resulting in a 20.8% reduction. Here, we can observe.

The analysis suggests that increasing the number of mini-sequences (M) can significantly reduce memory usage, especially for the LM-Head component, while having a minimal impact on execution time for longer sequences. Memory savings are more pronounced for the LM-Head than for the MLP. It is important to note that while MsT is highly beneficial for training with extremely long sequences, it may lead to performance degradation when applied to models with shorter sequences due to the overhead introduced by partitioning the input and the additional memory movement required for gradient accumulation. It can be easily observed from Table 13 that using M=32 mini-sequences increases the execution time from 0.01s (standard setting) to 0.11s for a sequence length of 1024, causing an 11x performance downgrade. Also, from Table 15, using M=8 mini-sequences increases the execution time from 0.05s (standard setting) to 0.12s for a sequence length of 1024, causing a 2x performance downgrade. The performance reduction is more pronounced for the LM-Head compared to the MLP. Fortunately, LM-head accounts for very little of the transformer's running time, which is smaller than MLP and much smaller than attention, so our technology will not affect the overall performance, even if it affects its module performance.

Table 16: MLP memory usage (in GB) for different sequence lengths and mini-sequence settings

| MLP memory | 1024 | 2048 | 4096 | 8192 | 20000 | 40000 | 80000 |
|---|---|---|---|---|---|---|---|
| standard | 0.93 | 1.09 | 1.39 | 2.11 | 4.18 | 7.69 | 14.72 |
| M=2 | 1.29 | 1.36 | 1.50 | 2.00 | 3.76 | 6.73 | 12.69 |
| M=4 | 1.32 | 1.41 | 1.61 | 2.00 | 3.49 | 6.21 | 11.66 |
| M=8 | 1.33 | 1.44 | 1.66 | 2.11 | 3.42 | 6.17 | 11.66 |

## G  Integrated with existing frameworks

MST's core idea is conceptually straightforward, primarily targeting MLP and LM-Head blocks. We offer two integration methods:

**Customized Hugging Face Transformer.**  This method involves directly modifying the Hugging Face Transformer library to incorporate MST functionality. By customizing the library, users can seamlessly integrate MST into their existing workflows that use Hugging Face Transformers. We made this method open-source on https://github.com/wdlctc/transformers.

To use the customized Hugging Face Transformer with MST, ML developer didn't change any line but install our customized transformers library with MsT:

```
import transformers
```

**Wrapper Mode.**  The Wrapper Mode provides a less invasive approach to integrating MST. This method involves creating a wrapper around existing model implementations, intercepting and modifying the forward and backward passes of the MLP and LM-Head blocks. We made this method open-source on https://github.com/wdlctc/mini-s.

To use the Wrapper Mode:

```
from mini-s import mst
model = mst(model)
```

**Conclusion.**  Both integration methods offer flexibility in adopting MST for long sequence training. The choice between Customized Hugging Face Transformer and Wrapper Mode depends on the specific requirements of the project, the level of integration desired, and the willingness to maintain custom libraries.

For users deeply invested in the Hugging Face ecosystem, the Customized Hugging Face Transformer method may be preferable. This method requires minimal changes to existing codebases which is already integrated with the Hugging Face ecosystem, and allows access to all Hugging Face features and optimizations. For those seeking a more flexible solution or working with multiple model implementations, the Wrapper Mode could be the better choice to used with customized codebase challenges.

