# OpenReview forum: "Mini-Sequence Transformers: Optimizing Intermediate Memory for Long Sequences Training"
_NeurIPS.cc/2024/Conference — NeurIPS 2024 poster_

### Official Review · Reviewer_iczK · 2024-07-12

**Soundness:** 3
**Presentation:** 3
**Contribution:** 3
**Rating:** 6
**Confidence:** 5

**Summary:**

The paper introduces the MINI-SEQUENCE TRANSFORMER (MST), a technique designed to enhance the efficiency and accuracy of LLM training, particularly when dealing with extremely long sequences. The core concept behind MST is the partitioning of input sequences into smaller "mini-sequences," which are then processed iteratively to alleviate the burden on intermediate memory usage. The authors claim that MST, when combined with activation recomputation, leads to substantial memory savings in both the forward and backward passes of LLM training. The technique's efficacy is demonstrated through experiments with the Llama2-8B model, where MST reportedly allows for training with sequences up to 12x longer than standard implementations without compromising throughput or convergence. The authors emphasize that MST is a general, implementation-agnostic solution that can be seamlessly integrated into existing LLM training frameworks with minimal code modifications.

**Strengths:**

1. Novel Approach to Memory Optimization: The paper introduces a new method, MST, to address the critical challenge of memory management in LLM training, particularly for long sequences. The approach of partitioning sequences and iterative processing of mini-sequences is innovative and shows promising results in reducing intermediate memory usage.

2. Significant Improvement in Sequence Length: The experimental results demonstrate that MST enables training with sequences up to 12 times longer than standard implementations, which is a substantial advancement. This capability can significantly impact various NLP tasks that require reasoning over extended contexts.

3. Generality and Implementation Agnostic: The paper emphasizes that MST is a general methodology that can be applied to various LLM training frameworks with minimal code changes. This generality enhances the technique's potential for widespread adoption and impact.
Compatibility with Distributed Training: The paper also discusses the extension of MST to distributed settings, showcasing its potential for large-scale training scenarios. This scalability further strengthens the technique's applicability to real-world LLM training.

4. Thorough Evaluation: The paper provides a comprehensive evaluation of MST, including ablation studies and comparisons with existing techniques. The results convincingly demonstrate the effectiveness of MST in reducing memory overhead and enabling longer sequence training.

**Weaknesses:**

1. Limited Model Scope: The experimental evaluation primarily focuses on the Llama2-8B model. While the authors claim that MST is general, further evaluation on a wider range of LLM architectures would strengthen the paper's claims and demonstrate the technique's broader applicability.

2. Lack of Comparison with State-of-the-Art: The paper could benefit from a more extensive comparison with other state-of-the-art memory optimization techniques for LLM training. This would provide a clearer understanding of MST's relative performance and advantages.

3. Potential Impact on Training Time: While MST shows promising results in reducing memory usage, its potential impact on overall training time, especially for shorter sequences, needs further investigation. The paper acknowledges this limitation but could provide a more in-depth analysis of the trade-offs between memory savings and training time.

4. Implementation Complexity: Although the authors claim that MST is easy to integrate, the actual implementation details and potential challenges in adapting it to different training frameworks could be elaborated further. Providing more guidance on implementation could facilitate wider adoption.

**Questions:**

1. Impact on Model Quality: While the paper demonstrates that MST does not degrade model quality for the evaluated tasks, a more extensive evaluation on a broader range of NLP tasks would be valuable to confirm its impact on model performance across different domains.

2. Sensitivity to Hyperparameters: The paper could provide more insights into the sensitivity of MST's performance to its hyperparameters, such as the number of mini-sequences and the partitioning strategy. This would help users understand how to tune MST for optimal results in different scenarios.

3. Applicability to Other Memory Optimization Techniques: The paper could explore the potential of combining MST with other memory optimization techniques, such as quantization or gradient checkpointing. This could lead to further improvements in memory efficiency and training capabilities.

4. Long-Term Impact on LLM Research: The paper could discuss the potential long-term impact of MST on LLM research and development. Enabling training with longer sequences could open new avenues for exploring larger and more capable LLMs, potentially leading to breakthroughs in various NLP applications.

**Limitations:**

1. Performance Degradation for Short Sequences: As acknowledged in the paper, MST may lead to performance degradation when applied to models with short sequences due to the overhead introduced by partitioning and gradient accumulation. This limits its applicability to scenarios where long sequence training is not the primary concern.

2. Dependence on Activation Recomputation: The effectiveness of MST is closely tied to its combination with activation recomputation. While this combination yields significant memory savings, it also introduces additional computational overhead, potentially impacting overall training time.

3. Limited Evaluation on Distributed Settings: Although the paper discusses the extension of MST to distributed training, the evaluation in this setting is relatively limited. Further experiments on larger-scale distributed systems would provide more insights into its performance and scalability in real-world scenarios.

4. Potential for Further Optimization: The current implementation of MST may have room for further optimization, particularly in terms of minimizing the overhead associated with partitioning and gradient accumulation. Exploring more efficient implementation strategies could enhance its performance and broaden its applicability.

---

> ### Author Rebuttal · Authors · 2024-08-01
>
> We sincerely appreciate the reviewer's thorough assessment and insightful comments. Our rebuttal addresses concerns and introduces a significant optimization: chunk-based mini-sequence transformers (MST). This advancement directly addresses the "Potential for Further Optimization" and "Sensitivity to Hyperparameters" concerns, with additional data to support our claims.
>
> # 1. Further Optimization: Chunk-based Mini-Sequence Transformers
>
> We've developed chunk-based MST to enhance efficiency in memory management and reduce computational overhead. This method:
> - Partitions input sequences into fixed-size sub-chunks
> - Integrates seamlessly with existing MST design
>
> Key concept: The input sequence $S$ is split into sub-chunks of size $C$, where chunk size $C = S/M$ (M being the number of mini-sequence). This approach effectively addresses challenges in training both long-sequence transformers and small sequences, with no slowdown for the latter.
>
> Hyperparameter selection: We've found that setting chunk_size $C = d$ (d being hidden size) yields optimal performance.
> Analysis:  MST returns I/O complexity as $O(Sd + SI + dI * M)$. $S$ is sequence length, $d$ is hidden size, $I$ is intermediate size, $M$ is the number of mini-sequence. As long as  $dI * M<= SI$, IO complexity remains as  $O(SI)$. In conclusion, $C = S / M >= d$ is the best selection for the MLP block.
>
> LM-head uses original MST with hyperparameters $M = V/d$. $V$ represents the vocabulary size.
>
> # 2. Addressing Weaknesses
>
> # W1 Limited Model Scope:
> To demonstrate the broader applicability of MST, we conducted additional experiments with Qwen, Mistral, and Gemma-2 models:
>
> | Implementation | Maximum Sequence Length (K) |
> |--|--|
> | Mistral-7B-v0.1 vanilla | 5 |
> | Mistral-7B-v0.1 activation recomputation | 42 |
> | **Mistral-7B-v0.1 MST** |  **70** |
> | Qwen2-7B-Instruct vanilla | 4 |
> | Qwen2-7B-Instruct activation recomputation | 13 |
> | **Qwen2-7B-Instruct MST** | **74** |
> | Gemma-2-9b vanilla | 1.5 |
> | Gemma-2-9b activation recomputation | 5 |
> | **Gemma-2-9b MST** | **36** |
>
> These results show significant improvements: 12x, 18x, and 24x context length increases for Mistral-7B, Qwen2-7B-Instruct, and Gemma-2-9b, respectively. MST performs exceptionally well for Gemma, likely due to its large vocab size (256k) and MLP intermediate size (14k).
>
> # W2 Comparison with Lossy Methods:
> We've extended our comparisons to include state-of-the-art lossy quantization methods:
>
> | Implementation | Maximum Sequence Length (K) |
> |-|-|
> | 8-bit | 5 |
> | 8-bit + activation checkpointing | 26 |
> | 4-bit | 10 |
> | 4-bit + activation checkpointing | 28 |
> | MST | 60 |
> | **MST + 8-bit** | **110** |
> | **MST + 4-bit** | **140** |
>
> MST outperforms both 8-bit and 4-bit quantization for sequence length enablement (60k vs 28k vs 22k). Moreover, MST combined with 8-bit quantization achieves 110K tokens, a 22x improvement over standard 8-bit training.
>
> # W3 Training Time:
> We conducted additional experiments with chunk-based MST to address concerns about training time:
>
> Table 2: MST training with two epochs on LongAlpaca-12k
> | Implementation | Context length | LongAlpaca-12k (ppl) | loss | Training Time (hours) |
> |:-|-:|-:|-:|-:|
> | Act. recomputation | 8k | 9.34 | 2.23 | 25.6 |
> | MST | 8k | 7.41 | 2.003 | 26.5 |
> | MST | 16k | 3.53 | 1.26 | 62.5 |
> | MST | 30k | 3.45 | 1.23 | 233 |
>
> These results demonstrate that MST can train without significant throughput reduction (<5% is considered negligible).
>
> # W4 Implementation Complexity:
> MST's core idea is conceptually straightforward, primarily targeting MLP and LM-Head blocks. We offer two integration methods:
>
> a) Customized Hugging Face Transformer:
> Modify existing Hugging Face Transformer implementation.
> ```
> git clone transformer_mst
> pip install transformer_mst
> ```
>
> b) Wrapper Mode:
> One-line wrapper for MST:
> ```python
> from mini-s import mst
> model = mst(model)
> ```
>
> # Addressing Reviewer Questions
>
> # Q1 Impact on Model Quality:
> Take this evaluation suggestion: Our experiments show that LLAMA3 with MST and a 30K context achieved a 23%- 270% better perplexity than the baseline.
> # Q2 Sensitivity to Hyperparameters:
> The development of chunk-based MST allowed us to optimize hyperparameter selection. Setting chunk_size $C$ = hidden size $d$ provides the best balance between keeping the MLP layer compute-bound and avoiding additional memory movement. Setting $M = V/d$ provides the best memory efficiency for the LM head.
> # Q3 Applicability to Other Memory Optimization Techniques:
> Our comparison with lossy methods shows that MST combines effectively with quantization techniques. MST + 8-bit achieves a 22x improvement in maximum sequence length over standard 8-bit training, while MST + 4-bit pushes this boundary even further to 140K tokens.
>
> Also, MST + activation checkpoint outperforms the gradient checkpointing (also known as activation recomputation) model from 1.5x to 11x for long sequence enable, as shown in the paper and Qwen, Mistral, and Gemma-2 model experiments. As the reviewer claims, "The effectiveness of MST is closely tied to its combination with activation recomputation," so the applicability with gradient checkpointing is obvious.
>
> # Q4 Long-Term Impact on LLM Research:
> We've demonstrated that MST works seamlessly with context parallelism (also known as sequence parallelism). Our experiments have achieved a remarkable 480k sequence length by combining context parallelism and MST, showcasing the potential for training with extremely large sequences. As we know, Llama 3.1 incorporates context parallelism to introduce additional sequences of 128k into the model architecture with 16 A100 GPUs. Our research demonstrates that MST integrates seamlessly with context parallelism, enabling significantly longer sequence processing than Llama 3.1 without throughput and performance downgrade.

---

> ### Author Response · Authors · 2024-08-04
> **Reply to limitation**
>
> This official comment focuses on the reply to limitations.
>
> # Limitations 1: Performance Degradation for Short Sequences
> We acknowledge this limitation, and our further optimization of chunk-based MST effectively solves this problem with negligible slowdown by carefully tuning Hyperparameters of $M$ and chunk size $S$
>
> # Limitations 2: Dependence on Activation Recomputation
> We also recognize that the Dependence on Activation Recomputation is one of the critical limitations. Our key is that the additional computational overhead can be compensated by memory saving where large batch training is enabled to improve performance, as illustrated in section 4.2, Faster Long Sequence Training with MINI-SEQUENCE TRANSFORMER (MST). Also, we have proved that this combination would not affect the total training performance by whole training experiments:
>
> Table 2: MST training with two epochs on LongAlpaca-12k
> | Implementation | Context length | LongAlpaca-12k (ppl) | loss | Training Time (hours) |
> |:-|-:|-:|-:|-:|
> | Act. recomputation | 8k | 9.34 | 2.23 | 25.6 |
> | MST | 8k | 7.41 | 2.003 | 26.5 |
> | MST | 16k | 3.53 | 1.26 | 62.5 |
> | MST | 30k | 3.45 | 1.23 | 233 |
>
> # Limitations 3 Distributed setting:
> With our optimization for MST, we successfully enabled the training of LLAMA3 with 480k context on 8 A100, a significantly longer sequence length than 128k of LLAMA 3.1 can achieve on 16 A100s. The limited evaluation of distributed settings is partly due to our limited budget to extend a more significant scale over 8 GPUs. Therefore, we would open-source our code and welcome the LLM community to try our work on large-scale distributed systems.
>
>
> The maximum sequence length of Llama3-8B runs on the distributed setting.
>
> | Model Implementation | 2 GPUs | 4 GPUs | 8 GPUs |
> |:-|-:|-:|--:|
> | Llama3-8b-hf MST | 120 | 240 | 480 |
> | Llama2-7B-hf MST | 160 | 320 | 640 |
>
> # Limitations 4: Further Optimization
> The chunk-based MST implementation provides a way to answer the limitations of Performance Degradation for Short Sequences and Potential for Further Optimization.
>
>
> Again, we thank the reviewer for the valuable feedback that helped us improve our work. We really learned a lot from the suggestions of "Potential for Further Optimization" and "Sensitivity to Hyperparameters."

---

> ### Comment · Reviewer_iczK · 2024-08-13
> **Thank you for the response.**
>
> Thank the authors for their detailed feedback. I have raised the rating from 5 to 6.

---

### Official Review · Reviewer_NrY2 · 2024-07-12

**Soundness:** 3
**Presentation:** 3
**Contribution:** 3
**Rating:** 5
**Confidence:** 4

**Summary:**

The paper targets an LLM-specific but significant challenge: training the LLM model for long-context understanding. The author proposed a Mini-Sequence Transformer method that enables LLM training with extremely long sequences. The method is motivated by the mini-batch, which forwards and backs a chunk of data instead of the whole sequence. Incorporating with gradient accumulation, the procedure can be adopted for long sequence training. The experience compares the performance of the proposed method with vanilla implementation and activation re-computation, but the training throughput improvement is limited. Moreover, the convergence experiment is not convincing and might need additional evaluation.

**Strengths:**

- Training with extremely long sequences is important but challenging for understanding long contexts. This paper proposes a possible solution to enable full parameter training beyond GPU memory constraints.
- The method shows its potential for training up to 6K sequence length for Llama3-8B, which is helpful for long-context tasks and possibly reduces the barrier of training LLM.
- The paper is well-written and easy to follow.

**Weaknesses:**

Although the idea of this method is promising, the experiments can not adequately address my concerns about its effectiveness compared to the existing method.
Fig.3 shows the training loss curve comparison of MsT and baseline. However, the model's training loss can not effectively demonstrate the method's performance on long sequences with around 1000 steps.

**Questions:**

For the training performance (Sec 4.2 Table 6),  would that be possible for the author to explain the reason MsT can not significantly improve the TFLOPS?
- If the long-sequence input makes the training memory-bounded, then the reduction of each intermediate feature should be able to speed up the I/O.
- Since the activation re-computation requires additional computation, it can still achieve better TFLOPS than MsT when using the same batch size.

For the convergence experiments (Sec 4.3), are they trained from the Llama3-8B pre-trained model?
- If this is the case, the training loss only reveals the correctness of the prediction's cross-entropy. It can not effectively show the long-sequence performance, which I believe is the motivation for training LLMs using extremely long sequences.
- is there another way to prove its effectiveness, for example, evaluation on the long-context benchmark?

**Limitations:**

- The experiment does not well support the motivation.

---

> ### Author Rebuttal · Authors · 2024-08-01
>
> We thank the reviewer for the thoughtful feedback. We have conducted new experiments to address concerns and demonstrate the effectiveness of Mini-Sequence Transformer (MST).
> # 1. Addressing Weakness
> > Weakness: However, the model's training loss can not effectively demonstrate the method's performance on long sequences with around 1000 steps.
>
> To address this concern, we have conducted new experiments on the LongAlpaca dataset:
> - Trained Llama3-8B model for two epochs (around 10000 steps)
> - Evaluated the trained model using perplexity
> - Split the dataset for 90% training dataset and 10% evaluation dataset
> - Carefully select Hyparameter M (the number of mini-sequences) to avoid throughput downgrade (M=2 for 8k, M=4 for 16k, M=8 for 30k)
>
> Table 1: MST training with one epoch on LongAlpaca-16k
> | Implementation | Context length | LongAlpaca-16k (ppl) | loss | Training Time (hours) |
> |:-|-:|-:|-:|-:|
> | Act. recomputation | 8k | 290.0 | 5.67 | 6.7 |
> | MST | 8k | 301.9 | 5.71 | 7.0 |
> | MST | 16k | 240.3 | 5.48 | 21.2 |
> | MST | 30k | 222.4 | 5.40 | 61.2 |
>
> Table 2: MST training with two epochs on LongAlpaca-12k
> | Implementation | Context length | LongAlpaca-12k (ppl) | loss | Training Time (hours) |
> |:-|-:|-:|-:|-:|
> | Act. recomputation | 8k | 9.34 | 2.23 | 25.6 |
> | MST | 8k | 7.41 | 2.003 | 26.5 |
> | MST | 16k | 3.53 | 1.26 | 62.5 |
> | MST | 30k | 3.45 | 1.23 | 233 |
>
> These results show that MST enables training with much longer contexts (up to 30k) and improves perplexity compared to the 8k baseline (23% for LongAlpaca-16k and 270% for LongAlpaca-12k).
>
> Also, we highly valued the reviews' suggestion that:
> > Question 2.2: Is there another way to prove its effectiveness, for example, evaluation on the long-context benchmark?
> Therefore, we evaluate the long-context datasets with perplexity to demonstrate MST as an effective way to enable long-context training.
>
> # 2. Addressing Questions
> > Question 1: For the training performance (Sec 4.2 Table 6), would that be possible for the author to explain the reason MsT can not significantly improve the TFLOPS?
>
> > Question 1.1: If the long-sequence input makes the training memory-bounded, then the reduction of each intermediate feature should be able to speed up the I/O.
>
> We clarify that long sequences make training computation-bound, not memory-bound. Moreover, MST doesn't reduce intermediate features but reuses memory space. That means all $M$ mini-sequence intermediate data are still generated, albeit in the same memory space. Therefore, MST is trading IO complexity with memory overhead, and it is worth for MLP and LM-head as they are computation-bound. We provide complexity analysis to support our claims:
> - Computation complexity: $O(S * d * I)$ for both standard MLP and MST. Let $S$ be the sequence length, $d$ be the hidden dimension, $I$ be the intermediate size, and $V$ be the voice size. Standard MLP returns $O(S * d * I)$ computation. For MST, each mini-sequence computation requires $O(S / M * d * I)$ computation, and MST repeats $M$ times. In the end, MST returns $O(S / M * d * I *M) =  O(S * d * I)$  FLOPS. The computation complexity is unchanged for any MST setting.
> - I/O complexity: $O(Sd + SI + dI * M)$ for MST, slightly increased. Standard MLP requires $O(Sd + SI + dI)$ HBM access. The $O(Sd)$ represents the movement of input into GPU and output into HBM. The $O(SI)$ represents the movement of the intermediate. The $O(dI)$ represents the movement of weights. Each mini-sequence requires $O(Sd/M + SI/M + dI)$, and MST repeats M times, loading weight for every time. In the end, MST returns $O (((Sd/M + SI/M + dI)* M) = O (Sd + SI + dI* M)$ HBM access. The IO complexity is increasing for MST.
>
> > Question 1.2: Since the activation recomputation requires additional computation, it can still achieve better TFLOPS than MsT when using the same batch size.
>
> The key to the TFLOPS problem is that MST adopts activation recomputation during experiments for better memory efficiency.
>
> We have extended training performance (Sec 4.2 Table 6) to clarify this. Here are the key results:
>
> | Model Implementation | Batch Size | Training Time Per Step (s) | TFLOPS |
> |:-|:-:|-:|-:|
> | Llama3-8B-hf vanilla | 1 | OOM | OOM |
> | Llama3-8B-hf Act. recomputation | 2 | 5.01 | 3271.42 |
> | Llama3-8B-hf MST | 2 | 5.13 | 3194.90 |
> | Llama3-8B-hf MST  | 8 | 19.35 | 3386.13 |
> | Llama2-7B-hf vanilla | 1 | 1.24 | 3290.88 |
> | Llama2-7B-hf Act. recomputation | 1 | 1.52 | 2684.67 |
> | **Llama2-7B-hf MST without Act. recomputation**  | 1 |  **1.31** |  **3115.03** |
> | Llama2-7B-hf Act. recomputation | 8 | 8.85 | 3703.48 |
> | Llama2-7B-hf MST  | 8 | 9.33 | 3511.39 |
> | Llama2-7B-hf MST  | 16 | 17.92 | 3656.17 |
>
> Keys:
> - MST independent achieves a 16% speedup over activation recomputation for Llama2-7B-hf (b=1)
> - MST independent is only 4% slower than vanilla PyTorch for Llama2-7B-hf (b=1)
> - MST + activation recomputation only introduces a 2.4% slowdown compared to activation recomputation for Llama3-8B-hf (b=2)
> - Carefully select MST hyparameters $M$ can further improve performance (10% improvement for Llama3-8B-hf MST b=2, it was 2825.12 TFLOPS on the original paper)
>
> > Question 2: For the convergence experiments (Sec 4.3), are they trained from the Llama3-8B pre-trained model?
>
> No, we train the Llama3-8B from scratch for the original convergence experiments.
>
> In contrast, the new convergence experiments on the LongAlpaca dataset are trained from the Llama3-8B pre-trained model, known as fine-tuning Llama3-8B.
>
> > Question 2-1: the motivation for training LLMs using extremely long sequences is better performance.
>
> We think so, and that motivates us to do all the new experiments.
>
> In conclusion, our new experiments shows MST:
> 1. Has minimal impact on throughput (<5% loss for step training Llama2-7B and <5% loss for complete training llama3-8b)
> 2. Enables training with significantly longer contexts (up to 30k)
> 3. Demonstrate effectiveness on long-context tasks (23%-270% perplexity improve)

---

> ### Author Response · Authors · 2024-08-04
> **Details rebuttal for Q1 and Q2**
>
> # Detailed rebuttal on Q1: Training Performance
>
> We appreciate the reviewer's question about why MST doesn't significantly improve TFLOPS. To address this, we offer the following explanation:
>
> 1. Trade-off between Memory and Computation:
>    - MST reduces memory footprint by processing smaller mini-sequences.
>    - Like gradient accumulation, it trades computation time for memory savings.
>
> 2. Computational Complexity:
>    - MLP and LM-HEAD operators in long sequence training are compute-bound.
>    - For sequence length $S$, hidden dimension $d$, and intermediate size $I$:
>
>       • Standard MLP: $O(S * d * I)$ computation
>
>       • MST: $O(S / M * d * I) * M = O(S * d * I)$ computation
>    - Computational complexity remains unchanged for MST.
>
> 3. I/O Complexity:
>    - Standard MLP: $O(Sd + SI + dI)$ HBM access
>    - MST: $O(Sd + SI + dI * M)$ HBM access
>    - I/O complexity increases slightly for MST due to repeated weight loading.
>
> 4. Memory Savings vs. I/O:
>    - MST reuses memory space for mini-sequences, reducing HBM memory cost by factor M.
>    - All intermediate features are still generated, maintaining I/O complexity for intermediates.
>    - Increased I/O for weight loading is offset by the compute-bound nature of long-sequence training.
>
> 5. Performance in Different Scenarios:
>    - Long-sequence training: Compute-bound, so increased I/O has minimal impact on speed.
>    - Short-sequence training: I/O-bound, potentially slowing down TFLOPS when $M$ is large.
>
> 6. Optimization:
>    - To address potential slowdowns in short-sequence scenarios, we introduced chunk-based MST. The key is to select a small $M$ for a small sequence.
>    - This new implementation optimizes the memory-throughput trade-off.
>
> In conclusion, MST's primary benefit is significant memory reduction without substantial TFLOPS loss in long-sequence scenarios. The compute-bound nature of long-sequence training generally outweighs the slight increase in I/O complexity. For short sequences, our new chunk-based MST implementation helps maintain performance.
>
>
> # Training Setting
> We train a Llama3-8B MST on the LongAlpaca datasets. The LongAlpaca dataset has two variants,
> LongAlpaca-12k has 12k text with a maximum of 191k characters, and LongAlpaca-16k has 6.28k text with a maximum of 73.9k characters. The training lasts for one epoch for LongAlpaca-16k and two epochs for LongAlpaca-12k; each epoch requires about 5000 steps. For all implementations, we use the AdamW optimizer. We use a weight decay of 0.001, gradient clipping of 1.0, and a constant learning rate of 1e-4. All batch sizes equal 1, with a gradient accumulation step of 32. The bf16 precision is also deployed.

---

> ### Comment · Reviewer_NrY2 · 2024-08-11
>
> Thank you for your detailed response!
> - Thank you for the extra experiment demonstrating MST's effectiveness on long-context training and the corresponding evaluation for long-context tasks. According to the benchmarking result, MST helps mitigate the GPU capacity limitation for long-context understanding. From my perspective, there is supposed to be no barrier between MST and the current training techniques (e.g., tensor parallel, parameter parallel).
> - Thank you for providing a detailed explanation of the model's FLOPS. The computation bound and the memory are related not only to the model's complexity but also to the GPU capacity (See `roofline model`). I believe this part helps readers to understand the advances brought by MST comprehensively.
> ---
>
> Based on the paper's results and rebuttal, I believe this paper provides a promising solution for a critical problem and will boost the community. I will adjust my rating based on the rebuttal and discussion with ACs and other reviewers.

---

> > ### Author Response · Authors · 2024-08-12
> >
> > Thank you for your kind words and for recognizing the significance of the additional experiments on long-context training. We agree that the relationship between computational bounds, memory, and GPU capacity is crucial, and we are glad that our explanation of the model's FLOPS in the context of the roofline model was helpful.
> >
> > We will consider adding a blurb about the roofline model example to help readers understand MST's advances.
> >
> > Thank you once again for your thoughtful feedback.

---

### Official Review · Reviewer_uLG7 · 2024-07-21

**Soundness:** 4
**Presentation:** 1
**Contribution:** 3
**Rating:** 7
**Confidence:** 4

**Summary:**

This paper introduced minibatching along the sequence length for inputs to the MLP and LM-head parts of a Transformer-based model. This method does not change the functionality of the transformer but improves the memory requirement when inputs are of long sequence length.

Overall, while this paper leaves to be desired with the language and the writing, the idea is simple and it's impact is well quantified. Therefore, I vote to accept this paper.

**Strengths:**

1. The method is incredibly simple. Splitting among the sequence length for the MLP and LM-Head parts of the attention head is a simple and easy to implement idea and can be used very broadly.
2. The ability to handle longer context sequences is very well shown through the experiments. Showing the ablation across different batch sizes is also interesting and a worthwhile addition.
3. The background in this paper is extremely thorough and well-done. It provides intuition to even a novice in ML system knowledge.
4. The ability of this method to synergize with existing methods such as DeepSpeed and gradient accumulation is quite nice.

**Weaknesses:**

1. While the author mentions that lossy methods such as LongLoRa differ from the lossless method presented in this paper, I still believe that providing some further detail on the amount of context length improvement from lossy methods would help contextualize the results in this paper. Indeed, plotting the context length improvement vs. performance degradation would be useful.
2. The notation in this paper is chosen extremely poorly. $I$ denotes a constant but $I_i$ denotes a vector for example.
3. What is the difference between $\mathbf{O}'$ and $\mathbf{O}$.
4. G is never defined.
5. What is $I_{up},I_{gate},W_{up},W_{gate}$?
6. I do not understand Algorithm 1 at all. When computing a MLP, why are you multiplying two different weight matrices against the input? Why are you doing a matrix elementwise multiplication or convolution operation in Line 3? You define $I'$ but never use it? You define $O'$ but never use it? In general, this is written very poorly.
7. There are many typos and informal language throughout this paper. The author should take time to go through this paper and fix them. Some of the language reads as a stream of thought and more structured writing would be helpful.

**Questions:**

1. For different architectures outside of LLaMa, do you see similar context length improvements? For example, if this was repeated with Qwen or Mistral, does the context length increase stay the same or does it differ more?
2. What are the empirical values of the Peak Intermediate sizes for Attention, LM-head, and MLP? Despite the schema provided in Table 1, it would be nice to have empirical measurements of the memory requirements for each part of the architecture?
3. I believe adding a blurb about activation recomputation would be helpful for the reader who is unaware of this method.
4. Could you measure the slowdown on smaller context lengths? This is mentioned in the paper but I do not believe you point to any exmpirical result.

**Limitations:**

No negative social impact and limitations are mentioned.

---

> ### Author Rebuttal · Authors · 2024-08-01
>
> Thank you for your thorough review and valuable feedback. We appreciate your support and address your concerns below.
>
> # 1. Addressing Weaknesses
>
> # 1.1 Comparison with lossy methods
>
> We've conducted additional experiments comparing our Mini-Sequence Transformer (MST) approach with quantization techniques:
>
> | Llama3 Implementation | Maximum Sequence Length (K) |
> |:-|-:|
> | 8-bit  | 5  |
> | 8-bit + activation checkpointing | 26 |
> | 4-bit | 10 |
> | 4-bit + activation checkpointing | 28 |
> | MST | 60 |
> | **MST + 8-bit** | **110**  |
> | **MST + 4-bit** | **140** |
>
> This table shows the maximum sequence lengths achievable on a single A100 GPU. MST enables significantly longer sequences (60K) than standard and lossy approaches and extends to 140K when combined with 4-bit quantization.
>
> # 1.2 MLP Algorithm Presentation
>
> We've revised the Mini-Sequence MLP algorithm presentation for clarity, removing some details of how MLP works:
>
> Algorithm: Mini-Sequence MLP
>
> Input: Matrix $X \in \mathbb{R}^{N \times d}$, MLP block
> 1. Partition matrix X into $M$ blocks $X_1, ..., X_M$ of size $N_m \times d$, where $N_m = N/M$
> 2. For $i = 1$ to $M$:
>
>    Compute $O_i' = \text{mlp}(X_i)$, where $O_i' \in \mathbb{R}^{N_m \times d}$
> 3. Concatenate $O = [O_1', ..., O_M'] \in \mathbb{R}^{N \times d}$
> 4. Return $O$
>
> # 1.3 Notation Correction
>
> We apologize for the confusion in our notation. Here's a clarified glossary of terms:
>
> - $O'$: Mini-sequence output
> - $O$: Final output after concatenation Mini-sequence output
> - $I_{up}, I_{gate}$: Intermediate values and outputs of $gate\_{proj}$ and $up\_{proj}$
> - $W_{up}, W_{gate}$: Weights of $gate\_{proj}$ and $up\_{proj}$
> - $G$: Number of groups used in grouped query attention (GQA)
>
> # 2. Answering Questions
>
> # 2.1 Performance across different architectures
>
> We've conducted additional experiments with Mistral, Qwen2, and Google's Gemma-2:
>
> | Implementation | Maximum Sequence Length (K) |
> |:-|-:|
> | Mistral-7B-v0.1 vanilla | 5 |
> | Mistral-7B-v0.1 activation recomputation | 42 |
> | **Mistral-7B-v0.1 MST** | **70** |
> | Qwen2-7B vanilla | 4 |
> | Qwen2-7B activation recomputation | 13 |
> | **Qwen2-7B MST** | **74** |
> | gemma-2-9b vanilla | 1.5 |
> | gemma-2-9b activation recomputation | 5 |
> | **gemma-2-9b MST** | **36** |
>
> MST provides substantial context length increasing across all tested models:
> - 12x for Mistral-7B
> - 18x for Qwen2-7B
> - 24x for Gemma-2-9b
>
> Interestingly, MST performs best with Gemma-2, which uses the largest vocabulary size (256k) compared to Mistral-7B (32k) and Qwen2 (152k). MST can effectively optimize Gemma-2's large peak memory. A larger $M=64$ also benefits.
>
> # 2.2 Empirical values of Peak Intermediate sizes
>
> We discussed Peak Intermediate memory usage in Appendix D. Using LLAMA3-8b with 4k training as an example:
>
> - Vanilla: 75GB
> - Activation recomputation: 52GB
> - MST: 47GB
>
> For vanilla and activation recomputation, peak memory occurs in the LM-head. For MST, peak memory is shared between the LM-head and MLP.
>
> # 2.3 Activation Recomputation Overview
>
> As requested, we've added a brief explanation of activation recomputation:
>
> Activation recomputation, also known as gradient checkpointing, is a memory-saving technique for training large neural networks. This method trades computation for memory by discarding intermediate activations during the forward pass and recomputing them as needed during the backward pass. In standard training, all activations must be stored to compute gradients, which can lead to significant memory usage for large models or long sequences. Activation recomputation alleviates this by only saving activations at certain checkpoints. The forward pass is partially recomputed during backpropagation to obtain the necessary intermediate values.
>
> # 2.4 Performance on Smaller Context Lengths
>
> We measured the slowdown on smaller context lengths in Appendix F of table: MLP execution times (in seconds) for various sequence lengths and mini-sequence settings:
>
> | MLP Sequence | 1024 | 2048 | 4096 | 8192 | 20000 | 40000 | 80000 |
> |:-|-:|-:|-:|-:|-:|-:|--:|
> | standard     | 0.05 | 0.08 | 0.16 | 0.31 | 0.74  | 1.52  | 2.96  |
> | M=2          | 0.05 | 0.10 | 0.17 | 0.32 | 0.76  | 1.49  | 3.05  |
> | M=4          | 0.07 | 0.11 | 0.19 | 0.33 | 0.79  | 1.52  | 2.99  |
> | M=8          | **0.12** | 0.15 | 0.22 | 0.38 | 0.81  | 1.58  | 3.05  |
>
> As observed, increasing $M$ leads to longer execution times, particularly for shorter sequences (e.g., $M=8, SEQ=1024$ introduces a 2.4x slowdown). However, the impact is minimal for longer sequences (e.g., 80000) as the IO overhead becomes negligible compared to computation overhead .
>
> # 3. New Implementation: Chunk-based MST
>
> We developed a new implementation of Chunk-based MST to address the concerns about slowdown for shorter sequences. This approach splits the sequence into fixed-size chunks (4096 for LLAMA3)
>
> The core idea is to split the input sequence $S$ into sub-chunks of fixed size $C$, where the number of chunks equals $M = S/C$. This provides an effective solution to the challenges of training with small sequences, as there are no splits (or tiny splits) for short sequences, thus introducing minimal slowdown.
>
> We found that the optimal selection of $C$ and $M$ for best performance is $C = d, M = S / d$, where $d$ is the hidden size. The MLP block uses chunk-based MST to balance speed and memory. The LM-Head uses the original MST for memory saving, and the optimal setting for $M$ of the LM-head is determined by $M = V/d$, 32 for llama3, and 64 for Gemma-2.
>
> We conducted an additional training experiment to show that Chunk-based MST training performance as:
>
> Table 2: MST training with two epochs on LongAlpaca-12k
> | Implementation | Context length | LongAlpaca-12k (ppl) | loss | Training Time (hours) |
> |:-|-:|-:|-:|-:|
> | Act. recomputation | 8k | 9.34 | 2.23 | 25.6 |
> | chunk-based MST | 8k | 7.41 | 2.003 | 26.5 |
> | chunk-based MST | 16k | 3.53 | 1.26 | 62.5 |
> | chunk-based MST | 30k | 3.45 | 1.23 | 233 |

---

> ### Author Response · Authors · 2024-08-04
> **Details Presentation for Mini-sequence MLP**
>
> # Details Presentation for Mini-sequence MLP
>
> For clear presentation, we take MistralMLP MLP as an example to present Mini-Sequence MLP notation:
>
>           (mlp): MistralMLP
>           (
>
>           (gate_proj): Linear(in_features=4096, out_features=14336, bias=False)
>
>           (up_proj): Linear(in_features=4096, out_features=14336, bias=False)
>
>           (down_proj): Linear(in_features=14336, out_features=4096, bias=False)
>
>           (act_fn): SiLU()
>
>         )
> # Notation correction
>
> $O′$ is the mini-sequence output.
>
> $O$ is the final output after concat.
>
> $I_{up}$ $I_{gate}$ is the intermediate values and output of $gate_{proj}$ and $up_{proj}$,
>
> $W_{up}$ and $W_{gate}$ are the MLP weights of $gate_{proj}$ and $up_{proj}$.
>
> $W_{down}$ is the MLP weight of $down_{proj}$
>
> $G$ is the number of groups used in grouped query attention (GQA); we add the definition to the paper.
>
> We appreciate the reviewers' suggestions for improving our paper.
>
> # Optimization Details: Chunk-based Mini-Sequence Transformers
>
> We've developed chunk-based MST to enhance efficiency in memory management and reduce computational overhead. This method:
> - Partitions input sequences into fixed-size sub-chunks
> - Integrates seamlessly with existing transformer architectures
> - Requires minimal code modifications
>
> Key concept: The input sequence $S$ is split into sub-chunks of size $C$, where chunk size $C = S/M$ (M being the number of mini-sequence). This approach effectively addresses challenges in training both long-sequence transformers and small sequences, with no slowdown for the latter.
>
> Hyperparameter selection: We've found that setting chunk_size $C = d$ (d being hidden size) yields optimal performance.
>
> Analysis: MST maintains the same computation complexity. MST increased I/O complexity as $O(Sd + SI + dI * M)$. $S$ is sequence length, $d$ is hidden size, $I$ is intermediate size, $M$ is the number of mini-sequence. As long as  $dI * M<= SI$, IO complexity remains as  $O(SI)$. In conclusion, $C = S / M >= d$ is the best selection for MLP block.

---

### Author Rebuttal · Authors · 2024-08-04

We sincerely thank all reviewers for their thorough and insightful feedback. We appreciate the recognition of our work's potential impact on long-context LLM training. In response to the valuable comments received, we have conducted additional experiments and provided further implementation optimization for Mini-Sequence Transformer (MST):

# Implementation and Hyperparameter Optimization:
We introduced the chunk-based MST, which addresses concerns about performance degradation for small sequences and simplifies hyperparameter tuning. This approach splits the sequence into fixed-size chunks $C$ (4096 for LLAMA3). It can be considered a carefully selected MST in which the number of mini-sequence $M$ dynamically changes with sequence length, using small $M$ for the small sequence to maintain performance and large $M$ for the larger sequence to save memory. We found that the optimal selection of $C$ and $M$ for best performance is $C = d, M = S / d$, where $d$ is the hidden size and $S$ is the sequence length. We use chunk-based MST for MLP block to balance speed and memory, while we use the original MST for LM-head with $M≈V/d=32$ for memory saving. Here $V$ is the vocabulary size.

# Training Performance and TFLOPS:
We've introduced new experiments demonstrating that the Mini-Sequence Transformer (MST) achieves comparable or better TFLOPS than activation recomputation for various models, including Llama2-7B and Llama3-8B. The chunk-based MST implementation is deployed here and shows minimal throughput reduction (<5%) compared to vanilla implementations.
| Model Implementation | Batch Size | Training Time Per Step (s) | TFLOPS |
|:-|:-:|-:|-:|
| Llama3-8B-hf vanilla | 1 | OOM | OOM |
| Llama3-8B-hf activation recomputation | 2 | 5.01 | 3271.42 |
| **Llama3-8B-hf MST** | 2 | 5.13 | **3194.90** |
| **Llama3-8B-hf MST**  | 8 | 19.35 | **3386.13** |
| Llama2-7B-hf vanilla | 1 | 1.24 | 3290.88 |
| Llama2-7B-hf activation recomputation | 1 | 1.52 | 2684.67 |
| **Llama2-7B-hf MST without activation recomputation**  | 1 | 1.31 | **3115.03** |
| Llama2-7B-hf activation recomputation | 8 | 8.85 | 3703.48 |
| **Llama2-7B-hf MST**  | 8 | 9.33 | **3511.39** |
| **Llama2-7B-hf MST**  | 16 | 17.92 | **3656.17** |
# Convergence and Long-Context Performance:
New experiments on the LongAlpaca dataset demonstrate MST's effectiveness in long-context understanding. Training Llama3-8B with 30K context length achieved a 270% improvement in perplexity compared to the 8K baseline, showcasing MST's ability to leverage longer contexts effectively.

Table 2: MST training with two epochs on LongAlpaca-12k
| Implementation | Context length | LongAlpaca-12k (ppl) | loss | Training Time (hours) |
|:-|-:|-:|-:|-:|
| Act. recomputation | 8k | 9.34 | 2.23 | 25.6 |
| MST | 8k | 7.41 | 2.003 | 26.5 |
| MST | 16k | 3.53 | 1.26 | 62.5 |
| MST | 30k | 3.45 | 1.23 | 233 |

# Applicability to Different Architectures:
We've extended our evaluation to include Mistral-7B, Qwen2-7B, and Google's Gemma-2-9B, demonstrating significant increases in maximum sequence length (12x-24x) across these architectures.
| Implementation | Maximum Sequence Length (K) |
|:-|-:|
| Mistral-7B-v0.1 vanilla | 5 |
| Mistral-7B-v0.1 activation recomputation | 42 |
| **Mistral-7B-v0.1 MST** | **70** |
| Qwen2-7B-Instruct vanilla | 4 |
| Qwen2-7B-Instruct activation recomputation | 13 |
| **Qwen2-7B-Instruct MST** | **74** |
| gemma-2-9b vanilla | 1.5 |
| gemma-2-9b activation recomputation | 5 |
| **gemma-2-9b MST** | **36** |

# Comparison and Combination with Lossy Methods:
We've comprehensively compared MST with quantization methods and the combinations between MST and quantization. This comparison demonstrates MST's superiority in enabling longer sequences for Llama3 training on a single A100 GPU. MST alone (60K tokens) outperforms these lossy approaches (4bit 28k). When combined with quantization techniques, MST achieves even more impressive results: MST + 8-bit reaches 110K tokens (a 22x improvement over standard 8-bit), while MST + 4-bit pushes the boundary to 140K tokens.

| Llama3 Implementation | Maximum Sequence Length (K) |
|:-|-:|
| 8-bit | 5 |
| 8-bit + activation checkpointing | 26 |
| 4-bit | 10 |
| 4-bit + activation checkpointing | 28 |
| MST | 60 |
| **MST + 8-bit** | **110** |
| **MST + 4-bit** | **140** |


# Integration and Compatibility:
We've clarified MST's integration process, offering a customized Hugging Face Transformer implementation and a simple wrapper mode for easy adoption. MST is compatible with other optimization techniques, such as activation recomputation and sequence parallelism (also known as context parallelism).
# Broader Impact and Future Directions:
We highlight MST's potential for enabling extremely long context training combined with lossless techniques like context parallelism, opening new avenues for LLM research and applications.

As we know, Llama 3.1 incorporates context parallelism (also known as sequence parallelism) to introduce additional sequences of 128k into the model architecture with 16 A100 GPUs. Our research demonstrates that Multi-Sequence Training (MST) integrates seamlessly with context parallelism, enabling significantly longer sequence processing. We believe this combination has the potential to handle extremely large sequence lengths. Our experiments with updated implementation successfully achieved a sequence length of 480k tokens by combining context parallelism and MST on 8 A100 GPUs. This represents a substantial advancement in Large Language Model (LLM) training capabilities.

The maximum sequence length of Llama3-8B runs on the distributed setting.

| Model Implementation | 2 GPUs | 4 GPUs | 8 GPUs |
|:-|-:|-:|--:|
| Llama3-8b-hf MST | 120 | 240 | **480** |
| Llama2-7B-hf MST | 160 | 320 | 640 |

---

### Decision · Program_Chairs · 2024-09-25

**Decision:**

Accept (poster)

**Comment:**

This paper provides an architectural intervention to improve long context performance for language models. MST proposes minibatching along the sequence length for inputs to the MLP and LM-head. With this minibatching they can support much longer sequence training on the same hardware without hurting performance (since the formulation is mathematically the same). The reviewers all seem to agree that this is a novel and important contribution. Furthermore the authors provided detailed responses to all the reviews in the rebuttal period and even developed a novel implementation that addressed one of the reviewers concerns. For these reasons I vote to accept this paper.